# Probing and Controlling Self-Reflection in Language Models

**Xudong Zhu**                                          *zhu.3944@osu.edu*
**Jiachen Jiang**                                       *jiang.2880@osu.edu*
**Mohammad Mahdi Khalili**                              *khaliligarekani.1@osu.edu*
**Zhihui Zhu**                                          *zhu.3440@osu.edu*
*Department of Computer Science and Engineering*
*The Ohio State University*
*Columbus, OH, USA*

**Reviewed on OpenReview:** *https://openreview.net/forum?id=AwVIfBZwyO*

## Abstract

Self-reflection—the ability of a large language model (LLM) to revisit, evaluate, and revise its own reasoning—has recently emerged as a powerful behavior enabled by reinforcement learning with verifiable rewards (RLVR). While self-reflection correlates with improved reasoning accuracy, its origin and underlying mechanisms remain poorly understood. In this work, *we first show that self-reflection is not exclusive to RLVR fine-tuned models: it already emerges, albeit rarely, in pretrained models.* To probe this latent ability, we introduce Reflection-Inducing Probing, a method that injects reflection-triggering reasoning traces from fine-tuned models into pretrained models. This intervention raises self-reflection frequency of Qwen2.5 from 0.6% to 38.9%, revealing a hidden capacity for reflection. Moreover, our analysis of internal representations shows that both pretrained and fine-tuned models maintain hidden states that distinctly separate self-reflective from non-reflective contexts. Leveraging this observation, *we then construct a self-reflection vector, a direction in activation space associated with self-reflective reasoning.* By manipulating this vector, we enable bidirectional control over the self-reflective behavior for both pretrained and fine-tuned models. Experiments across multiple reasoning benchmarks show that enhancing these vectors improves reasoning performance by up to 13.7%, while suppressing them reduces computational cost, providing a flexible mechanism to navigate the trade-off between reasoning quality and efficiency without requiring additional training. Our findings further our understanding of self-reflection and support a growing body of work showing that understanding model internals can enable precise behavioral control. Our code is publicly available at https://github.com/xzAscC/ProbingReflection.

## 1 Introduction

Reinforcement Learning with Verifiable Rewards (RLVR) has emerged as a powerful technique for enhancing the reasoning abilities of large language models (LLMs), enabling learning from outcome-level feedback across diverse tasks (Xu et al., 2025; WANG et al., 2025; Mroueh, 2025). In a nutshell, RLVR optimizes for end-task success, enabling models to autonomously explore and discover novel reasoning strategies at scale without relying on explicit supervised signals from human expertise (Zhao et al., 2025b; Ferrag et al., 2025; Su et al., 2025). Notably, it has been reported (Guo et al., 2025; Liu et al., 2025c; Zeng et al., 2025) that such training induces new emergent behaviors, such as self-reflection—the ability of a model to revisit, evaluate, and revise its prior outputs. For instance, the DeepSeek-R1 report (Guo et al., 2025) highlights that RLVR-trained models often generate tokens such as "wait", interpreted as signals of internal deliberation or critique.

At the same time, a growing body of work raises an important question: does RLVR truly incentivize new reasoning capabilities beyond those already present in the base pretrained model, or does it primarily elicit and amplify latent behaviors through training and decoding strategies (Yue et al., 2025)? This ongoing

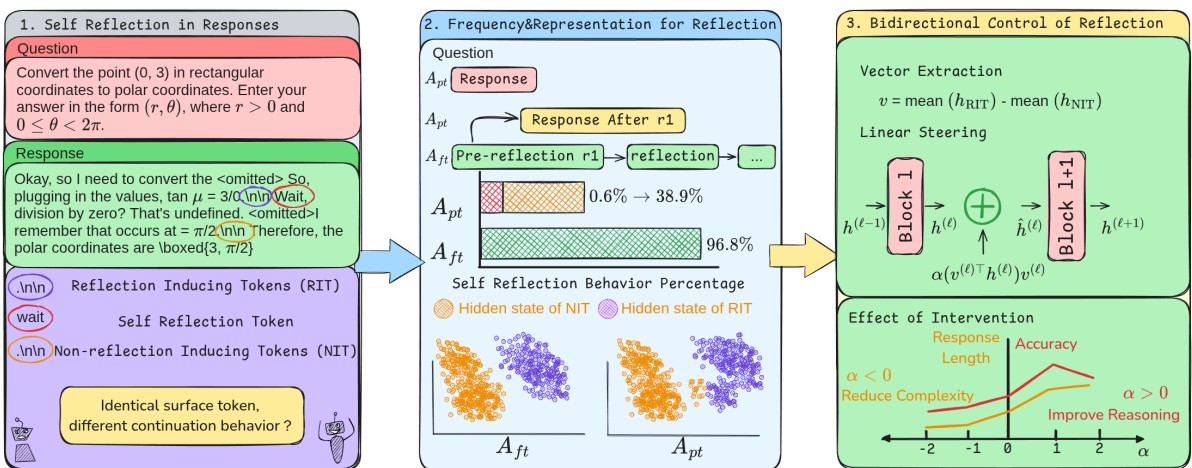

Figure 1: The overall pipeline of our method. **Left:** We identify self-reflection behavior in LLMs by using LLM-as-a-judge to locate self-reflection tokens (e.g., *wait*), and extract the corresponding Reflection-Inducing Tokens (RIT) and Non-Reflection-Inducing Tokens (NIT) from their context (see 4.2 for definitions). **Middle:** Hidden state analysis reveals that both $\mathcal{A}_{\text{pt}}$ and $\mathcal{A}_{\text{ft}}$ exhibit clear geometric separation between reflection and non-reflection states, indicating that $\mathcal{A}_{\text{pt}}$ retains latent self-reflection capacity despite its low surface-level reflection rate (0.6%). Transplanting pre-reflection responses from $\mathcal{A}_{\text{ft}}$ into $\mathcal{A}_{\text{pt}}$ confirms this capacity, boosting its self-reflection rate to 38.9%. **Right:** Leveraging RIT and NIT hidden states collected from both models, we construct a self-reflection vector and apply linear steering. Positive steering strength improves task performance, while negative steering yields more concise responses, demonstrating bidirectional control over the self-reflection behavior.

debate calls for a more principled understanding of what is genuinely learned versus what is revealed. In this work, we focus on one such behavior—self-reflection—as a concrete lens to study this question.

Despite these observations of self-reflection emerging under RLVR, it remains unclear whether such behaviors are genuinely induced by RLVR or arise from latent capabilities in the base model, and more fundamentally, why and how such reflective behaviors emerge. Answering this question is crucial for understanding the foundations of reasoning in LLMs and guiding future methods for enhancing their performance. Moreover, empirical studies have shown that reflection correlates with more accurate and robust reasoning (Zuo et al., 2025; Yue et al., 2025; Liu et al., 2024a), and that prompting models to explicitly "wait" or reflect can further improve performance with test-time computing. However, this benefit may come at a cost: reflection can increase inference time, introduce unnecessary verbosity, and reduce computational efficiency (Yang et al., 2025b; Renze & Guven, 2024; Sui et al., 2025). These findings highlight not only the need to understand self-reflection in LLMs, but also the importance of controlling it to balance reasoning quality and efficiency. These gaps motivate this work to study the following research questions:

*Is self-reflection a novel behavior induced by RLVR, or does it already emerge during pretraining? Can we control self-reflection in LLMs to balance performance and computational efficiency?*

**Contribution** In this work, we provide affirmative answers to both questions. First, we compare the reasoning behaviors of pretrained models and fine-tuned models (either via RLVR or distillation), and verify that self-reflection is already present in the pretrained model, albeit at a much lower frequency. Next, we analyze the hidden representations associated with reflective versus non-reflective reasoning, and find that they exhibit distinct activation patterns. Furthermore, we show that the degree of self-reflection can be modulated by a single direction in the representation space. Our contribution can be summarized as follows.

- **Self-Reflection already emerges during pretraining:** We demonstrate that self-reflection capabilities naturally exist in pretrained models and are not solely artifacts of RLVR. [1] However, the frequency of such behavior is extremely low—for example, only 0.6% as shown in Figure 1. To isolate the model's capacity for self-reflection from its general reasoning ability, we propose a method, called *reflection-inducing probing*, that inserts reasoning traces—specifically, those that trigger self-reflection in a fine-tuned reasoning model—into the input of the pretrained model, and then measures whether the latter produces reflection in response. Using reflection-inducing probing, we observe that the pretrained model exhibits reflection with a frequency of 38.9%, significantly higher than the baseline of 0.6%, though still lower than the fine-tuned model (which is almost 100%). Through comparative analysis of hidden representations, we show that pretrained models maintain internal structures that distinguish reflective behavior from non-reflective contexts—similar to fine-tuned models—further suggesting that pretrained models already possess self-reflection capabilities.
- **The degrees of self-reflection can be modulated by a single direction:** Motivated by the separability of reflective and non-reflective contexts in the hidden representation space, we use the method of difference-in-means (Rimsky et al., 2024) to construct a *self-reflection direction*, enabling control over self-reflection behavior for both pretrained and fine-tuned models. Our experiments demonstrate that this control mechanism offers a tunable trade-off between accuracy and efficiency: enhancing reflection improves accuracy on benchmarks by up to 13.7%, while suppressing it reduces output length by up to 32% without significant performance degradation. We further show that this direction transfers robustly across diverse tasks—including mathematical and scientific reasoning—highlighting its universality as a shared, task-agnostic mechanism.

## 2 Related Work

### 2.1 Features as Directions

Extracting feature directions, often derived from contrastive pairs of inputs, is an established technique for analyzing and manipulating neural network representations (Rimsky et al., 2024; Burns et al., 2022; Zou et al., 2023). It is widely recognized that adding such feature vectors to the model's residual stream can modify its behavior, although the optimal intervention points and specific methodologies remain areas of active research (Von Rütte et al., 2024; Jorgensen et al., 2023).

Several studies suggest that activation-space directions can capture semantic features more effectively than neurons (Geiger et al., 2024; Park et al., 2023; Bolukbasi et al., 2016). These directions can be identified using methods ranging from sparse autoencoders (Huben et al., 2024) to simple contrastive estimators such as Difference-in-Means, which has shown stronger performance in model steering (Arditi et al., 2024; Wu et al., 2025). The broader linear representation hypothesis has also motivated successful interventions for behavioral control (Shao et al., 2023; Belrose et al., 2023; Feng et al., 2025; Zhu et al., 2025). Nevertheless, we do not assume that all behaviors are single-directional; linear directions should be viewed as a useful first-order approximation, while some behaviors may require richer structures such as subspaces or manifolds.

### 2.2 Self-Reflection in Language Models

The concept of self-reflection in language models has gained increasing attention as a mechanism for improving reasoning quality and alignment. Recent studies (Lightman et al., 2024; Madaan et al., 2023; Puerto et al., 2024; Zelikman et al., 2022; Lightman et al., 2023; Li et al., 2025) have explored how prompting models to generate intermediate reflections, critiques, or alternative solutions can improve final outputs in tasks such as math problem solving, programming, and factual reasoning. While many of these techniques are implemented at the prompting level or through chain-of-thought scaffolding, they suggest that self-reflection is a powerful tool for enhancing reasoning. Notably, these methods often induce substantial increases in generation length and latency, raising questions about the trade-off between deliberation and efficiency (Yang

---

[1]However, whether self-reflection should be interpreted as genuine metacognitive capability requires further investigation and is beyond the scope of this paper. Our claims throughout are therefore about reflective behavior and its internal representations, rather than about metacognition per se.

et al., 2025b; Yi & Wang, 2025; Chen et al., 2024; Team et al., 2025). This places reflection control within the broader literature on test-time computation (Ai et al., 2025; Huang et al., 2025). Concurrent work such as (Wang et al., 2025) studies over-reflection from the surface-token perspective, showing that suppressing explicit markers such as "wait" can reduce unnecessary deliberation.

Related work further shows that RLVR can enhance reasoning performance by encouraging reflection through outcome-based feedback (Shah et al., 2025; Shinn et al., 2023; Wang et al., 2025; Xu et al., 2025). Empirically, models such as DeepSeek-R1 exhibit improved reasoning ability with emergent self-reflective behaviors (Guo et al., 2025; Liu et al., 2025b). However, concurrent studies suggest that RLVR may amplify latent capabilities already acquired during pretraining, rather than introducing fundamentally new ones (Shah et al., 2025; Yue et al., 2025). Our work complements this view by showing that self-reflection is a broadly distributed latent feature of LLMs, already encoded in hidden states even without explicit reflection training, and can be selectively enhanced or suppressed through simple vector interventions. This enables lightweight control of reflective behavior, including the mitigation of over-reflection, without additional training.

## 3 Preliminary

### 3.1 Transformer Layer

Decoder-only transformers (Liu et al., 2018) map input tokens $\mathbf{t} = (t_1, t_2, \ldots, t_n) \in \mathcal{V}^n$ to output probability distributions over the vocabulary $\mathcal{V}$. Let $\mathbf{h}_i^{(\ell)} \in \mathbb{R}^d$ denote the residual stream activation (also referred to as the hidden state) of the $i$-th token at the $\ell$-th layer, where $d$ is the dimensionality of the hidden state. Each of the $L$ transformer layers applies a sequence of attention and MLP to update the residual stream:

$$\widetilde{\mathbf{h}}_i^{(\ell)} = \mathbf{h}_i^{(\ell)} + \text{Attn}^{(\ell)}(\mathbf{h}_{1:i}^{(\ell)}), \quad \mathbf{h}_i^{(\ell+1)} = \widetilde{\mathbf{h}}_i^{(\ell)} + \text{MLP}^{(\ell)}(\tilde{\mathbf{h}}_i^{(\ell)}). \tag{1}$$

The final hidden state is then projected to a probability distribution over the vocabulary $\mathcal{V}$ using an unembedding matrix followed by a softmax function.

### 3.2 Self-Reflection

Recent work has shown that large language models (LLMs), even when pretrained purely on next-token prediction, demonstrate surprising levels of reasoning ability (Mondorf & Plank, 2024; Liu et al., 2024b; Wang et al., 2023). However, this capability can be significantly enhanced through fine-tuning on reasoning tasks using either reinforcement learning with verifiable rewards (RLVR) or supervised learning with distilled responses from reasoning models trained from RLVR (Liu et al., 2025a; Wang et al., 2024; Zhao et al., 2025a). We denote the pretrained model as $\mathcal{A}_{\text{pt}}$, and its fine-tuned variant as $\mathcal{A}_{\text{ft}}$.

A notable emergent behavior observed in fine-tuned models $\mathcal{A}_{\text{ft}}$ is self-reflection—the model's ability to internally evaluate, critique, or revise its own reasoning process. Unlike standard reasoning, which involves generating a direct solution to a task, self-reflection introduces an intermediate step in which the model pauses or backtracks to reconsider its prior outputs (Li et al., 2024; Liu et al., 2025c; Yeo et al., 2025). Throughout this paper, we use self-reflection as a behavioral label for this revision pattern, without committing to a stronger meta-cognitive interpretation of the underlying mechanism.

Importantly, self-reflection is not limited to any specific model architecture, observed across both proprietary models (Jaech et al., 2024) and open-source systems (Guo et al., 2025; OLMo et al., 2024; Yang et al., 2025a), indicating that it may be a general emergent property of optimizing for complex reasoning objectives. Although phrases such as "wait" are often treated as markers of self-reflection, recent work shows that a model may generate such expressions as superficial discourse patterns, without actually revising its prior reasoning (Wang et al., 2025). For this reason, we do not define self-reflection using a fixed set of tokens alone. Instead, we use an LLM-as-a-judge (Gu et al., 2024) to evaluate complete responses and determine whether they exhibit self-reflective behavior. When self-reflection is present, we further distinguish between responses that merely contain surface-level reflection and those that demonstrate genuine reconsideration of prior reasoning. This procedure allows us to move beyond token matching and obtain a broader grounded characterization of self-reflection. Further details are provided in Section 6.3.

# 4 Self-Reflection Already Emerges During Pretraining

In this section, we conduct a systematic analysis of self-reflection behavior in pretrained models $\mathcal{A}_{\tt pt}$ and fine-tuned reasoning models $\mathcal{A}_{\tt ft}$. Using the MATH500 dataset (Hendrycks et al., 2021) as our evaluation benchmark, we compare their reasoning trajectories. We first show that self-reflection emerges naturally in pretrained models, although it appears at a substantially lower frequency than in post-trained reasoning models. We then analyze hidden-state representations and find that even pretrained models implicitly encode and distinguish between self-reflective and non-self-reflective states, despite rarely expressing reflective behavior explicitly in their outputs.

## 4.1 Probing Self-Reflection in Pretrained Models

To investigate whether self-reflection emerges intrinsically in $\mathcal{A}_{\tt pt}$, rather than being solely a byproduct of fine-tuning strategies such as RLVR, we examined the behavior of $\mathcal{A}_{\tt pt}$ on mathematical reasoning tasks using the MATH500 benchmark. Figure 1 (middle) highlights the contrast in self-reflection frequency between $\mathcal{A}_{\tt pt}$ and $\mathcal{A}_{\tt ft}$ models, alongside the clear geometric separation of their hidden states into reflection and non-reflection clusters. Even in the absence of task-specific supervision, $\mathcal{A}_{\tt pt}$ exhibited spontaneous self-reflective behavior in a small but non-negligible fraction of cases—approximately 0.6%. These instances are characterized by explicit reconsideration or revision of prior reasoning steps. For details on how we identify self-reflection instances, please see Section 6.3. While the self-reflection tokens differ somewhat from those typically observed in RLVR-trained models, their reflective nature is still discernible. These findings suggest that self-reflection is not solely acquired through fine-tuning, but rather emerges as a latent capability within the $\mathcal{A}_{\tt pt}$, one that is infrequently activated but nonetheless present.

**Reflection-Inducing Probing by Injecting CoT from $\mathcal{A}_{\tt ft}$ into $\mathcal{A}_{\tt pt}$** However, the extremely low frequency of self-reflection in $A_{\tt pt}$ makes it challenging to analyze systematically and to develop methods (which will be studied in the next section) for controlling such behavior. To address this challenge, we propose a probing method, termed *reflection-inducing probing*, that isolates the model's capacity for self-reflection from its general reasoning ability. The key idea is to decouple reasoning competence from reflective behavior by inserting reasoning traces generated by the fine-tuned model $\mathcal{A}_{\tt ft}$ into the input of the pretrained model $\mathcal{A}_{\tt pt}$, and then measuring whether $\mathcal{A}_{\tt pt}$ generates reflection in response.

Formally, given a question $q$, we use the fine-tuned model $\mathcal{A}_{\tt ft}$ to generate a sequence of reasoning tokens:

$$\mathcal{A}_{\tt ft}(q) = (\underbrace{r_1}_{\text{pre-reflection}}, \underbrace{\tt reflection}_{\text{explicit signal}}, \underbrace{r_2}_{\text{post-reflection}})$$

where $r_1$ denotes the initial chain-of-thought leading up to an explicit reflection token (e.g., "wait"), and $r_2$ represents the revised or continued reasoning after reflection. We then construct a new prompt by inserting $r_1$ (the pre-reflection reasoning) into the input of $\mathcal{A}_{\tt pt}$, and evaluate whether $\mathcal{A}_{\tt pt}$ independently produces a reflection token at the appropriate point. This setup ensures that both models operate on similar reasoning contexts, eliminating confounding differences in reasoning capability. By comparing the frequency and consistency of self-reflection under this controlled setting, we can more directly assess whether reflective behavior is present in the pretrained model and to what extent it is amplified by fine-tuning. The frequency of generating reflection with reflection-inducing probing is reported in Figure 1.

**Self-Reflection emerges naturally in pretrained models albeit with much lower frequency** $\mathcal{A}_{\tt pt}$ exhibits clear self-reflective behavior in 38.9% of these cases, a dramatic increase from its baseline rate. This differential response demonstrates that while $\mathcal{A}_{\tt pt}$ rarely produces overt reflection markers in standard contexts, it possesses latent self-reflection capabilities that can be activated by appropriate contextual triggers. These findings strongly suggest that self-reflection mechanisms are encoded during pretraining, rather than being exclusively developed through reinforcement learning. With 38.9% self-reflection cases, the subsequent section analyzes the hidden state representations of these self-reflective behaviors to provide further insights.

### 4.2 Hidden State Representations of Self-Reflection

To further investigate the emergence of self-reflection, we analyze the internal representations of the model when it decides to generate reflection versus when it does not. Specifically, we focus on the hidden states associated with reasoning tokens that immediately precede the generation of a reflection token (e.g., "wait"), and compare them to those that do not lead to reflection.

Since both the pretrained model $\mathcal{A}_{\mathtt{pt}}$ and the fine-tuned one $\mathcal{A}_{\mathtt{ft}}$ exhibit self-reflection behaviors, we use $\mathcal{A}$ to denote a generic model (either pretrained or fine-tuned), which will be specified in context. Given a question $q$, suppose the model $\mathcal{A}$ generates a sequence of reasoning tokens that includes self-reflection. Let $r = \mathcal{A}(q) = (r_1, \mathtt{reflection}, r_2)$, where $r_1$ precedes the reflection token and $r_2$ follows it. Due to the auto-regressive nature of transformer models, the information from the question $q$ and the reasoning tokens $r_1$ is aggregated into the hidden representation of the final token in $r_1$, which is then used by the last layer to predict the next token—the reflection token. For convenience, we refer to the final token in $r_1$ as a *reflection-inducing token (RIT)*, though its hidden state captures information from the entire preceding context $(q, r_1)$. Reflection-inducing tokens often coincide with sentence-final punctuation (e.g., ".", "!", or closing brackets) or specific markers such as "But". Now with a slight abuse of notation, let $h^{(\ell)}_{\text{reflection-inducing}}(q, r)$ denote the $\ell$-th layer hidden state of a model at $\mathcal{A}$ corresponding to the reflection-inducing token. We collect all such hidden states from model outputs that contain reflection tokens into the following set

$$\mathbb{H}^{(\ell)}_{\mathtt{reflect}} = \left\{ h^{(\ell)}_{\text{reflection-inducing}}(q, r) \in \mathbb{R}^d \right\}. \tag{2}$$

To study the properties of hidden states associated with self-reflection, we contrast this set with representations from cases where the model *does not* generate reflection. Specifically, to eliminate the confounding effect of token surface form, we extract hidden states from tokens that share the same form as reflection-inducing tokens (e.g., sentence-final punctuation), but which do not lead to self-reflection in the subsequent responses (within 20 tokens in the experiments). For notational convenience, we refer to these as *non-reflection-inducing tokens*, and denote their corresponding hidden states as $h^{(\ell)}_{\text{non-reflection-inducing (NIT)}}(q, r)$. We collect such non-reflection-inducing into the following set

$$\mathbb{H}^{(\ell)}_{\mathtt{non-reflect}} = \left\{ h^{(\ell)}_{\text{non-reflection-inducing}}(q, r) \in \mathbb{R}^d \right\}. \tag{3}$$

This design ensures a fair comparison by controlling for the surface form of the reflection-inducing token, ensuring that any differences in hidden representations are attributable to the model's decision to reflect.

For $\mathcal{A}_{\mathtt{pt}}$ models, which rarely generate self-reflective outputs, we use the method of reflection-inducing probing by injecting CoT from $\mathcal{A}_{\mathtt{ft}}$ into $\mathcal{A}_{\mathtt{pt}}$ to elicit reflective behavior. To visualize these high-dimensional representations, we employ principal component analysis (PCA) for dimensionality reduction, projecting the hidden states into a 2D space. Figure 1 presents the visualizations for the 15th layer (out of 28 total layers) for both models, with more layers presented in Appendix B.

Our analysis reveals a pattern: **both models show clear separation between self-reflection and nonself-reflection states**. While this separation is expected in $\mathcal{A}_{\mathtt{ft}}$, which was explicitly trained to exhibit self-reflective behavior, the equally distinct clustering in $\mathcal{A}_{\mathtt{pt}}$ is remarkable. Despite rarely generating explicit self-reflection tokens in its outputs, $\mathcal{A}_{\mathtt{pt}}$ maintains internal representations that clearly distinguish between self-reflective and nonself-reflective contexts. This finding provides strong evidence that self-reflection capabilities develop during pretraining, with models encoding these patterns in their hidden state representations even when they rarely manifest in generated text. We will exploit this internal structure to develop methods for controlling self-reflection in the next section.

## 5 Controlling Self-Reflection in Language Models

In this section, we introduce our approach for identifying and manipulating self-reflection vectors in LLMs. Building on our finding in the last section that hidden representations distinctly separate self-reflective from non-reflective contexts, we construct self-reflection vectors, directions in activation space associated with

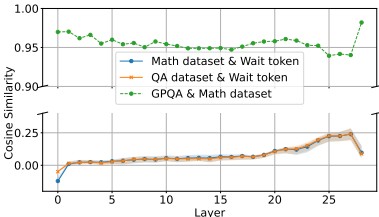 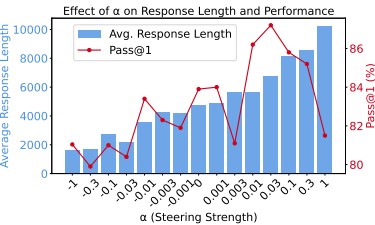 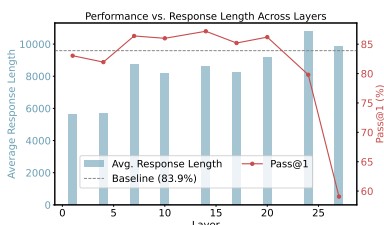

Figure 2: Analysis of self-reflection vectors in DeepSeek-R1-1.5B. **Left**: cosine similarity between reflection vectors across datasets and their similarity to the "wait" token. **Middle**: Effect of steering strength $\alpha$ on performance and response length on MATH500 injected at layer 14. **Right:** Effect of injecting the self-reflection vector at different layers of DeepSeek-R1 1.5B ($\alpha = 0.03$) on Pass@1 and response length for MATH-500.

self-reflective reasoning. We then demonstrate how these vectors can be used to bidirectionally control self-reflection behavior, either enhancing it to improve reasoning accuracy or suppressing it to reduce computational overhead. Through extensive evaluation across multiple mathematical reasoning benchmarks, we show that our method significantly outperforms strong baselines while offering flexible control over the performance-efficiency trade-off. Finally, we examine the cross-domain transferability of these vectors, revealing their potential as universal controls for self-reflection across diverse reasoning tasks.

## 5.1 Extract Self-Reflection Vectors

To identify the self-reflection vector in the residual stream activations, we compute the difference between the activations of self-reflective and nonself-reflective contexts. This technique, known as difference-in-means, effectively isolates key feature directions, as demonstrated in prior work (Rimsky et al., 2024; Arditi et al., 2024; Wu et al., 2025), motivating our application of this approach to the self-reflection domain. As in Section 4, we focus on the hidden state of reflection-inducing token, positing that this state encodes the model's transition into self-reflection reasoning.

For each layer $\ell \in \{1, \ldots, L\}$, we compute the mean hidden states in the reflection set $\mathbb{H}^{(\ell)}_{\texttt{reflect}}$ and non-reflection set $\mathbb{H}^{(\ell)}_{\texttt{non-reflect}}$, using 100 examples for each set, as

$$\boldsymbol{\mu}^{(\ell)}_{\texttt{reflect}} = \text{mean}(\mathbb{H}^{(\ell)}_{\texttt{reflect}}), \quad \boldsymbol{\mu}^{(\ell)}_{\texttt{non-reflect}} = \text{mean}(\mathbb{H}^{(\ell)}_{\texttt{non-reflect}}), \tag{4}$$

and then construct the self-reflection vector as the difference-in-means vector $\boldsymbol{v}^{(\ell)} = \boldsymbol{\mu}^{(\ell)}_{\texttt{reflect}} - \boldsymbol{\mu}^{(\ell)}_{\texttt{non-reflect}}$, which captures both the direction along which self-reflective and nonself-reflective activations diverge, and the magnitude of that divergence.

## 5.2 Model Interventions for Controlling Trade-off between Reasoning and Efficiency

To actively modulate a model's tendency to reflect, motivated by the linear representation hypothesis and prior work (Arditi et al., 2024), we apply simple linear interventions based on the self-reflection vector $\boldsymbol{v}^{(\ell)}$ from the $\ell$-th layer, which is expected to capture the direction in representation space most associated with self-reflection. Specifically, we modify each residual stream $\boldsymbol{h}^{(\ell)}$ at the $\ell$-th layer in (1) according to

$$\widehat{\boldsymbol{h}}^{(\ell)} = \boldsymbol{h}^{(\ell)} + \alpha \boldsymbol{v}^{(\ell)} \left\langle \boldsymbol{h}^{(\ell)}, \boldsymbol{v}^{(\ell)} \right\rangle, \tag{5}$$

where $\widehat{\boldsymbol{h}}^{(\ell)}$ replaces $\boldsymbol{h}^{(\ell)}$ as the input to the next layer, and the $\alpha$ controls the intervention strength. Here, $\left\langle \boldsymbol{h}^{(\ell)}, \boldsymbol{v}^{(\ell)} \right\rangle$ is a scalar projection coefficient measuring how strongly the current hidden state aligns with the self-reflection direction. When $\alpha > 0$, the self-reflection behavior is enhanced; when $\alpha < 0$, it is suppressed.

| Model | Method | MATH-500 | | AIME 2024 | | GPQA Diamond | |
|---|---|---|---|---|---|---|---|
| | | Pass@1 ↑ | LEN ↓ | Pass@1 ↑ | LEN ↓ | Pass@1 ↑ | LEN ↓ |
| $\mathcal{A}_{\texttt{ft}}$ Models | | | | | | | |
| DeepSeek-R1-1.5B | Vanilla | 84.1 | 4755 | 29.2 | 6118 | 14.0 | 4250 |
| | BF | 85.5 | 10122 | 30.0 | 8986 | 14.8 | 8210 |
| | SR Enhanced | **87.4** | 9458 | **33.5** | 8132 | **18.9** | 7496 |
| | SR Suppressed | 83.2 | **3716** | 27.3 | **5229** | 13.8 | **3795** |
| DeepSeek-R1-7B | Vanilla | 92.7 | 3585 | 55.8 | 4558 | 27.1 | 3696 |
| | BF | 93.1 | 7111 | 54.5 | 9629 | 32.0 | 5802 |
| | SR Enhanced | **93.5** | 8959 | **58.2** | 5684 | **34.6** | 6513 |
| | SR Suppressed | 91.2 | **2439** | 52.9 | **3319** | 26.5 | **3120** |
| OLMo2-13B | Vanilla | 41.8 | 3075 | 7.2 | 3753 | 15.2 | 2645 |
| | BF | 43.2 | 4891 | 10.2 | 4894 | 17.0 | 3714 |
| | SR Enhanced | **47.9** | 4574 | **11.7** | 4941 | **19.1** | 3490 |
| | SR Suppressed | 40.8 | **2106** | 6.5 | **2656** | 13.7 | **1375** |
| OLMo3-7B-Think | Vanilla | 94.5 | 7351 | 61.6 | 7212 | 47.6 | 8301 |
| | BF | 95.0 | 8251 | 62.3 | 9856 | 48.0 | 10543 |
| | SR Enhanced | **95.4** | 7690 | **65.3** | 9653 | **51.2** | 10015 |
| | SR Suppressed | 94.2 | **4822** | 61.3 | **5201** | 47.0 | **5544** |
| $\mathcal{A}_{\texttt{pt}}$ Models | | | | | | | |
| Qwen2.5-1.5B | Vanilla | $27.5 \pm 1.3$ | $\mathbf{1515 \pm 242}$ | $0.2 \pm 0.0$ | $\mathbf{544 \pm 98}$ | $5.7 \pm 0.7$ | $\mathbf{702 \pm 40}$ |
| | BF | $29.6 \pm 1.4$ | $3993 \pm 312$ | $0.1 \pm 0.2$ | $4517 \pm 428$ | $5.0 \pm 0.8$ | $1389 \pm 205$ |
| | SR Enhanced | $\mathbf{36.9 \pm 2.3}$ | $1836 \pm 174$ | $\mathbf{6.1 \pm 1.7}$ | $2525 \pm 233$ | $\mathbf{6.8 \pm 0.9}$ | $814 \pm 96$ |
| Qwen2.5-7B | Vanilla | $44.8 \pm 1.9$ | $\mathbf{1294 \pm 15}$ | $3.9 \pm 0.4$ | $\mathbf{2285 \pm 297}$ | $16.2 \pm 0.6$ | $\mathbf{598 \pm 143}$ |
| | BF | $46.0 \pm 1.1$ | $3986 \pm 351$ | $5.3 \pm 0.9$ | $4526 \pm 401$ | $\mathbf{16.9 \pm 1.2}$ | $1639 \pm 188$ |
| | SR Enhanced | $\mathbf{56.6 \pm 2.7}$ | $2671 \pm 214$ | $\mathbf{16.1 \pm 2.4}$ | $2941 \pm 265$ | $14.8 \pm 1.5$ | $1816 \pm 203$ |
| Llama 3.1-8B | Vanilla | $44.0 \pm 2.4$ | $\mathbf{1613 \pm 189}$ | $3.5 \pm 0.4$ | $\mathbf{1653 \pm 124}$ | $30.9 \pm 1.2$ | $\mathbf{1914 \pm 351}$ |
| | BF | $40.1 \pm 1.8$ | $3771 \pm 294$ | $2.9 \pm 0.8$ | $3912 \pm 377$ | $26.1 \pm 1.9$ | $5031 \pm 462$ |
| | SR Enhanced | $\mathbf{57.7 \pm 2.1}$ | $2887 \pm 241$ | $\mathbf{16.5 \pm 2.6}$ | $2974 \pm 286$ | $\mathbf{34.1 \pm 1.7}$ | $3995 \pm 355$ |

Table 1: Performance on mathematical and scientific reasoning benchmarks across $\mathcal{A}_{\texttt{ft}}$ and $\mathcal{A}_{\texttt{pt}}$ models. We compare **Vanilla** (no intervention), **BF** (budget forcing via "wait" token insertion), and our method **SR Enhanced/SR Suppressed** (applying positive or negative $\alpha$ to amplify or suppress self-reflection during inference). For $\mathcal{A}_{\texttt{pt}}$ models, we report only **SR Enhanced**, since self-reflection triggers it only rarely. Pass@1 denotes accuracy (higher is better), and LEN denotes average generation length (lower is better). We use 10% of the data as a validation set to select the optimal $\alpha$ in Eq. 5.

**Ablation study on $\alpha$** To illustrate the effect of the linear intervention method for controlling self-reflection, we conduct an ablation study by varying the self-reflection steering strength $\alpha$ from $-1.0$ to $1.0$, injecting the reflection vector at layer 14 in DeepSeek-R1-1.5B on the MATH-500 benchmark. The result is shown in Figure 2. Negative $\alpha$ values shorten responses, reducing average token length, while preserving accuracy. In contrast, positive $\alpha$ both lengthens responses, and boosts performance, peaking at $\alpha=0.03$ with a 12% performance gain in Pass@1, before declining at larger values due to over-reflection. This clear trade-off underscores $\alpha$ as a practical knob for balancing verbosity against reasoning depth. Further ablation detail on the effect of the injection layer is provided in Section 6.5.

## 5.3 Experimental Results

We evaluate our self-reflection control mechanism along three complementary dimensions: **reasoning ability**, **reasoning efficiency**, and **reasoning quality preservation**. For reasoning ability and efficiency, we report Pass@1 and average response length on two mathematical reasoning benchmarks, MATH-500 (Hendrycks et al., 2021) and AIME 2024, as well as one scientific QA benchmark, GPQA Diamond (Rein et al., 2024). To assess whether steering self-reflection alters the semantic quality of the reasoning process itself, we additionally evaluate the generated chains using ROSCOE-SS (Golovneva et al., 2023), which measures semantic agreement across reasoning steps using `facebook/roscoe-512-roberta-base` embeddings.

| Setting | DeepSeek-R1-1.5B | DeepSeek-R1-7B | OLMo2-13B | OLMo3-7B-Think | Qwen2.5-1.5B | Qwen2.5-7B | Llama 3.1-8B |
|---|---|---|---|---|---|---|---|
| Vanilla | 0.562 | **0.589** | **0.537** | **0.594** | 0.518 | 0.541 | **0.576** |
| SR Enhanced | 0.558 | 0.586 | 0.534 | 0.589 | **0.523** | **0.549** | 0.561 |
| SR Suppressed | **0.565** | 0.577 | 0.533 | 0.591 | \ | \ | \ |

Table 2: ROSCOE-SS scores on MATH500 across all models. Higher is better. We compare **Vanilla** and our method **SR Enhanced/SR Suppressed**. For $\mathcal{A}_{\text{pt}}$ models, we report only **SR Enhanced**, since self-reflection is triggered only rarely and suppression is therefore less informative.

Experiments are conducted on both $\mathcal{A}_{\text{pt}}$ and $\mathcal{A}_{\text{ft}}$. For $\mathcal{A}_{\text{pt}}$ models, we use Qwen2.5-1.5B, Qwen2.5-7B and Llama 3.1 8B (Grattafiori et al., 2024; Yang et al., 2024). For $\mathcal{A}_{\text{ft}}$, we evaluate DeepSeek-R1 1.5B and 7B, OLMo-2-13B-Instruct and OLMo3-7B-Think (Guo et al., 2025; OLMo et al., 2024; Olmo et al., 2025). All evaluations use greedy decoding. For $\mathcal{A}_{\text{pt}}$ models, we report the mean and standard deviation across steering vectors constructed from CoT traces generated by four different $\mathcal{A}_{\text{ft}}$ models. For $\mathcal{A}_{\text{ft}}$ models, we use a fixed validation-selected steering configuration and report point estimates.

We compare three inference strategies: **Vanilla**, which uses standard decoding without intervention; **BF** (Budget Forcing), which encourages reflection by appending a "wait" token to generations shorter than 8196 tokens (Guo et al., 2025; Muennighoff et al., 2025); and **SR Enhanced/SR Suppressed**, our method that perturbs hidden states using self-reflection vectors scaled by a coefficient $\alpha$ (positive for enhancement, negative for suppression). Since $\mathcal{A}_{\text{ft}}$ models can frequently trigger self-reflection, we evaluate both enhancement and suppression for them; in contrast, $\mathcal{A}_{\text{pt}}$ models exhibit such behavior only rarely, so we report SR enhancement only. We use 10% of the data as a validation set to select the optimal $\alpha$ in Eq. 5. Table 1 reports reasoning ability and efficiency, while Table 2 evaluates whether these interventions preserve the semantic quality of the reasoning chain.

**Key Findings.** Our results highlight four major insights. First, SR Enhancement improves reasoning performance across most evaluated datasets and model sizes. For example, Qwen2.5 7B's performance on MATH-500 jumps by 11.8 percentage points (from 44.8% to 56.6%), and DeepSeek-R1 variants enjoy similar boosts when employing reflection-enhanced decoding. Furthermore, this performance gain is also accompanied by a noticeable increase in response length, suggesting that longer responses can be beneficial when tackling more challenging reasoning tasks.

Second, SR Suppression offers fine-grained control over computational cost. It consistently reduces output length while preserving most of the model's accuracy. Notably, DeepSeek-R1-7B reduces average token length from 3585 to 2439 on MATH-500 with only a minor drop in Pass@1, which remains above 91%.

Third, these effects demonstrate strong generalizability across training paradigms and model families. The observed improvements hold for both $\mathcal{A}_{\text{ft}}$ models (e.g., DeepSeek-R1, OLMO) and $\mathcal{A}_{\text{pt}}$ models (e.g., Qwen2.5, Llama3.1). One exception is Qwen2.5-7B on GPQA Diamond, where SR Enhanced slightly hurts performance. We hypothesize that self-reflection is most useful when the dominant errors are self-check-correctable, such as arithmetic mistakes, local inconsistencies, or flawed intermediate derivations. For GPQA Diamond, however, Qwen2.5-7B may be limited more by missing or uncertain domain knowledge, so stronger reflection can lead to longer deliberation over flawed premises rather than improved answers.

Fourth, self-reflection steering does not show a clear degradation in ROSCOE-SS. Across model families, the scores remain broadly comparable to the vanilla setting, suggesting that our intervention can modulate reflective behavior without obviously degrading the generated reasoning traces under this metric. Since ROSCOE-SS is only a proxy for reasoning-trace quality, we treat this result as a complementary diagnostic.

Building on this generalizability, we emphasize the practical value of inference-time control over latent self-reflection dynamics. Unlike rigid interventions such as budget forcing, our method affords semantically grounded, continuous modulation of a model's internal self-reflection, enabling a tunable trade-off between performance and efficiency, especially in resource-constrained settings.

# 6 Further Analyses of Self-Reflection Vectors

## 6.1 Transferability of Self-Reflection Vectors

To investigate the transferability of self-reflection vectors across different reasoning domains, we evaluated whether vectors extracted from the GPQA Diamond dataset could be effectively transferred to mathematical reasoning tasks in MATH500. We compute the cosine similarity between self-reflection vectors extracted from different domains (MATH500 and GPQA Diamond), and between these vectors and the embedding of the token "wait" in DeepSeek-R1-Distill-Qwen-1.5B. For tokenizers containing multiple sub-word tokens for "wait", we report the average cosine similarity along with its variance. The results are plotted in Figure 2.

| Model | Setting | PASS@1 | Len |
|-------|---------|--------|-----|
| 1.5B | SR Enhanced | 86.4 | 7781 |
| | SR Suppressed | 82.6 | 3684 |
| 7B | SR Enhanced | 93.0 | 6007 |
| | SR Suppressed | 91.0 | 2992 |

Table 3: Performance on MATH500 when applying self-reflection vectors extracted from GPQA Diamond to DeepSeek-R1 models.

Our analysis revealed remarkable consistency in the neural signatures of self-reflection across these distinct domains. Specifically, vectors extracted from GPQA and MATH500 exhibit high cosine similarity, suggesting that the internal representation of reflective states is largely domain-invariant. Notably, these self-reflection vectors are substantially different from the embedding of the token "wait", indicating that they encode deeper semantic properties of reflective behavior rather than surface-level cues.

Further, we evaluate the performance of self-reflection vectors derived from GPQA-Diamond on MATH500 using our proposed intervention method, SR-Enhanced/Suppressed, and present the results in Table 3. Notably, we observe similar performance gains to those seen with in-domain self-reflection vectors, as reported in Table 1. This cross-domain transfer demonstrates that the reflective mechanism captures a generalizable cognitive pattern rather than being confined to task-specific reasoning strategies. Together, these findings suggest that LLMs develop a unified internal representation of self-reflection, one that can be leveraged across tasks without the need for domain-specific fine-tuning.

## 6.2 Comparison of Different Methods

To validate our choice of the difference-in-means approach for generating steering vectors, we conduct a comparative analysis against alternative feature-extraction techniques. We consider four variants: (i) a difference-in-means direction with the projected update used in the main paper, (ii) a difference-in-means direction with additive steering, (iii) a standard PCA direction, and (iv) contrastive PCA (Abid et al., 2018). All methods are evaluated under identical settings.

Table 4: Comparison of feature-extraction methods and steering methods for constructing the self-reflection vector on DeepSeek-R1 1.5B.

| | SR Enhanced | | SR Suppressed | |
|--------|-------------|-------|---------------|-------|
| **Method** | **PASS@1 ↑** | **LEN ↓** | **PASS@1 ↑** | **LEN ↓** |
| Difference-in-means | **87.4** | **9458** | **83.2** | **3716** |
| ↪ Additive Steering | 86.0 | 9845 | 83.0 | 3639 |
| PCA | 84.3 | 10074 | 79.1 | 3912 |
| Contrastive PCA | 84.7 | 9876 | 80.6 | 4078 |

The evaluation was performed under identical intervention settings for both the enhancement and suppression of the target behavior. The quantitative results are shown in Table 4. The data indicate that the difference-in-means approach achieves the highest task performance in both scenarios, yielding superior PASS@1 scores while maintaining reasonable lengths.

This empirical finding aligns with insights from recent work in mechanistic interpretability (Wu et al., 2025). These studies suggest that for the specific goal of causal steering of model behavior, simple directional vectors derived from differences often outperform other techniques, thereby motivating our choice of the difference-in-means approach.

| Model | Onset Token | Evidence Quote |
|-------|-------------|----------------|
| DeepSeek-R1-1.5B | `wait` | "Wait, that cannot be right because I double-counted ..." |
| DeepSeek-R1-7B | `wait` | "Wait, let me re-check the sign here ..." |
| OLMo2-13B | `But` | "But this step assumes independence, which is not valid ..." |
| OLMo3-7B-Think | `hold` | "Hold on, I should recompute from the definition ..." |
| Qwen2.5-1.5B | `wait` | "Wait, I need to verify whether the carry term is correct ..." |
| Qwen2.5-7B | `hmm` | "Hmm, that earlier conclusion does not follow from the equation ..." |
| Llama 3.1-8B | `wait` | "Wait, I made an invalid simplification in the previous step ..." |

Table 5: Self-reflection examples for each model. For each model, we report one reflection token identified by the judge together with a short supporting quote from a response labeled as `GENUINE_SELF_REFLECTION`.

### 6.3 Identifying Self-Reflection Instances

To identify self-reflection in language model outputs, we use an LLM-as-a-judge procedure instead of relying solely on a fixed set of keywords. We define a response as exhibiting genuine self-reflection only when the model explicitly revisits its earlier reasoning and performs a meaningful check, correction, backtrack, revision, or re-derivation. This criterion is intentionally stricter than keyword matching: discourse markers such as `wait`, `hold on`, or `let me think` are not considered sufficient unless they are followed by substantive reconsideration of prior reasoning.

Given a question-response pair, the judge assigns one of three labels: `NO_REFLECTION`, `SURFACE_REFLECTION`, or `GENUINE_SELF_REFLECTION`. For responses labeled as `GENUINE_SELF_REFLECTION`, the judge further identifies all self-reflection occurrences in the response. For each occurrence, the judge returns (i) the reflection phrase, (ii) the corresponding reflection token, and (iii) a short evidence quote. This allows us to recover the full set of self-reflection points within a generation, rather than restricting analysis to only the first reflective turn. The full judge prompt is provided in Appendix C.

We apply this procedure to model outputs on the MATH500 benchmark, using a maximum generation length of 8196 tokens. Final labels and self-reflection tokens are determined by the judge rather than by the presence of any single keyword. To provide qualitative intuition, Table 5 presents one representative example for each model, together with a reflection token identified by the judge and a short supporting quote. As a complementary analysis, we also examine the surface tokens associated with self-reflection in Table 6. We find that the specific tokens are not identical across models, but they exhibit substantial overlap. In particular, markers such as `wait`, and

| Model | Top-5 Reflection Onset Tokens |
|-------|-------------------------------|
| DeepSeek-R1-1.5B | `wait`, `but`, `hold`, `hmm`, `actually` |
| DeepSeek-R1-7B | `wait`, `but`, `hold`, `so`, `actually` |
| OLMo2-13B | `but`, `wait`, `however`, `hold`, `actually` |
| OLMo3-7B-Think | `hold`, `wait`, `but`, `hmm`, `actually` |
| Qwen2.5-1.5B | `wait`, `hmm`, `but`, `hold`, `actually` |
| Qwen2.5-7B | `hmm`, `wait`, `but`, `actually`, `hold` |
| Llama 3.1-8B | `wait`, `but`, `hold`, `actually`, `hmm` |

Table 6: Top-5 most frequent reflection onset tokens for each model, as identified by the judge on responses labeled `GENUINE_SELF_REFLECTION`.

`actually` recur across multiple model families, suggesting that genuine reflective revision is often expressed through a partially shared set of surface cues, despite model-specific variation.

### 6.4 Linear Separability of Self-Reflection States

To quantify whether self-reflection states are linearly encoded in model representations, we train a linear SVM at each layer to distinguish reflection-inducing tokens from non–reflection-inducing tokens in Qwen2.5-1.5B and DeepSeek-R1-Distill-Qwen-1.5B. The classifier is trained on 10% of the hidden states and evaluated on the remaining 90%.

As shown in Figure 3, test accuracy exceeds 99% across all layers for both models. This result indicates that reflection-inducing and non–reflection-inducing tokens are almost perfectly linearly separable in the original

hidden-state space, suggesting that self-reflection is represented as a robust latent direction rather than a weak or diffuse signal.

Notably, this linear structure also appears in Qwen2.5-1.5B, which produces very few explicit self-reflection tokens. This finding supports our claim that self-reflection is encoded internally even in models that rarely express it overtly, and is therefore not tied only to surface markers or a specific training recipe.

This linear separability also helps explain the effectiveness of our steering method: the intervention acts along an existing representational axis that already distinguishes reflective from non-reflective states, allowing it to amplify or suppress self-reflection by leveraging an emergent internal structure. Complementary PCA visualizations in Appendix B provide a qualitative view of the same geometric pattern.

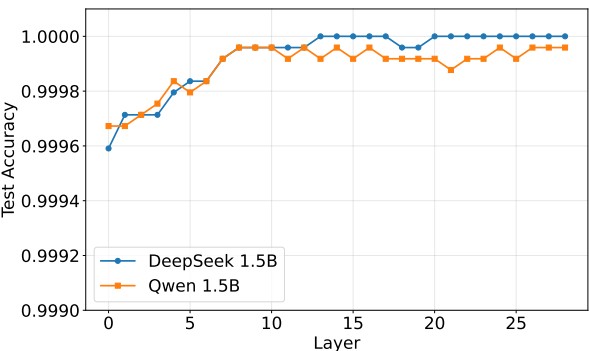

Figure 3: Linear-probe test accuracy of a linear SVM trained to distinguish reflection-inducing tokens from non–reflection-inducing tokens at each layer of DeepSeek-R1-Distill-Qwen-1.5B and Qwen2.5-1.5B. Accuracy exceeds 99% across all layers for both models, indicating near-perfect linear separability.

### 6.5 Ablation Study

We determine the injection configuration using a two-stage selection procedure. First, for each candidate layer $\ell$, we conduct a grid search over the steering coefficient $\alpha \in [-1.0, 1.0]$ to identify the value that yields the best validation performance. This search covers both positive and negative directions, corresponding respectively to enhancing and suppressing self-reflective behavior. Second, after identifying effective scales, we compare candidate injection layers to determine where the model is most receptive to intervention.

**Ablation on Steering Strength $\alpha$.** We first examine the effect of steering magnitude by varying $\alpha$ from $-1.0$ to $1.0$ while fixing the injection layer to layer 14 of DeepSeek-R1-1.5B. The results are shown in Figure 2. Overall, the relationship between $\alpha$ and performance is clearly asymmetric: moderate positive values are beneficial, whereas overly large magnitudes in either direction are harmful.

For negative $\alpha$, the intervention suppresses the self-reflection direction. In this regime, response length decreases substantially, indicating reduced deliberative reasoning. Notably, moderate negative values shorten outputs without causing a large drop in Pass@1, suggesting that part of the model's reflective behavior may be verbose rather than essential. For positive $\alpha$, the model is pushed further along the self-reflection direction. Small positive values improve Pass@1 and generally increase response length, indicating that stronger reflective reasoning can help the model revise faulty intermediate steps. However, this benefit is not monotonic. Performance peaks around $\alpha = 0.03$, after which larger positive values begin to hurt both answer quality and efficiency. Taken together, these results suggest that self-reflection is most useful when it is moderately amplified rather than maximized.

**Effect of Injection Layer.** We then isolate the role of layer choice by fixing $\alpha = 0.03$ and injecting the self-reflection vector at each layer of DeepSeek-R1-1.5B on MATH-500 (Figure 2). We observe a clear layer dependence: performance improves most strongly when the intervention is applied at middle layers, with layer 14 giving the best overall result.

This pattern suggests that self-reflection is most effectively modulated at intermediate stages of computation. When injected into early layers, the steering signal has to pass through many subsequent transformations, which can dilute or overwrite its effect before it influences the final reasoning trajectory. As a result, early-layer interventions produce only limited gains. In contrast, injecting into very late layers places the perturbation close to token generation, where it is less able to reshape the latent reasoning process and more likely to act as a superficial bias on next-token prediction.

The strong performance of middle-layer injection is consistent with the view that these layers encode higher-level reasoning dynamics in a form that is both semantically meaningful and still causally effective for downstream computation (Jiang et al., 2025; Skean et al., 2025). Intervening at this stage allows the model to revise or strengthen its internal reflective proc ess before the final answer is produced.

## 7 Conclusion

In this paper, we demonstrated that self-reflection in large language models is an emergent capability that develops during pretraining rather than being uniquely induced by reinforcement learning techniques. Through contrastive analysis of hidden state representations, we revealed that even models with minimal explicit reflection behavior maintain internal neural signatures that distinguish self-reflective contexts. By exploiting these representations, we developed an intervention method that enables bidirectional control over self-reflection, providing a flexible mechanism to navigate the performance-efficiency trade-off without requiring additional training. This work primarily focuses on understanding the internal representations underlying self-reflection and leveraging them for controllable behavior. There remain several important directions for future work. Our method requires users to decide in advance whether to enhance or suppress reflection, rather than adapting to task difficulty automatically. It also requires white-box access to internal activations, limiting its use for closed-source or API-only models. Moreover, our experiments are limited to models up to 13B parameters, leaving it open whether reflection vectors behave similarly in larger models. We leave adaptive control, restricted-access settings, and larger-scale evaluation to future work.

## Acknowledgements

We acknowledge support from the National Science Foundation under grants IIS-2312840, IIS-2402952, IIS-2301599, CMMI-2301601, and DMS-2529302. We thank members of the OSU CSE community for valuable discussions and feedback. We thank the Ohio Supercomputer Center for providing the computational resources.

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

## A  Implementation Details

For the $\mathcal{A}_{\text{ft}}$ models, we adopted a specialized prompting strategy that incorporates the explicit token `<think>` to elicit self-reflective reasoning and promote internal deliberation.

In contrast, for the $\mathcal{A}_{\text{pt}}$ models, we employed the same prompt template but omitted the `<think>` token. This design allowed us to isolate and assess the specific influence of `<think>` on eliciting reflective behaviors and its downstream impact, as Qwen models do not natively rely on such explicit triggers.

Unless otherwise noted, we set the maximum generation length to 32,784 tokens to support complex multi-step reasoning and avoid premature truncation. All experiments were conducted on a computing cluster equipped with 8 NVIDIA A5000 GPUs.

## B  PCA Visualization of Self-Reflection States

To complement the quantitative linear-probe results in the main text, we apply PCA to hidden states extracted from Qwen2.5-1.5B and DeepSeek-R1-Distill-Qwen-1.5B. Figure 1 shows the PCA projection of the 15th-layer representations, while Figures 4 and 5 provide additional layer-wise visualizations for both models.

Across layers, reflection-inducing and non–reflection-inducing tokens form two elongated clusters that become increasingly separated along the first principal component. In very early layers there is still some overlap, but from middle layers onward a clear margin emerges, and by the deepest layers the two groups are well separated in both model families.

These PCA visualizations are consistent with the linear-probe results reported in the main text, providing a low-dimensional illustration of the same underlying geometric structure: self-reflection corresponds to a direction in representation space that becomes progressively more organized across depth.

## C  LLM-as-a-Judge Pipeline for Identifying Self-Reflection

**A three-way taxonomy of reflective behavior.** Throughout this pipeline we distinguish three types of responses. Genuine self-reflection requires the model to explicitly revisit its earlier reasoning and meaningfully re-evaluate, or correct it, for example, identifying an error in a previous step, or re-deriving and amending an intermediate result. Surface reflection contains reflection-like discourse markers such as `wait`, but no marker is ever followed by a substantive revision: every previously stated conclusion remains unchanged. No reflection contains neither reflective markers nor reflective behavior. The decisive boundary is thus behavioral rather than lexical: what separates genuine from surface reflection is not the presence of hesitation markers, but whether prior reasoning is actually revised.

**Judge validation.**

To assess whether the LLM judge can reliably distinguish among the three reflection categories, we construct a manually curated validation set containing 25 examples from each category. The genuine-reflection subset consists of responses selected from `RLHFlow/Deepseek-MATH500-Test` in which the model revisits and corrects an erroneous intermediate reasoning step. The no-reflection subset consists of the first 25 examples from `HuggingFaceFW/fineweb-edu`, which are ordinary web-text passages containing no reflective behavior. The surface-reflection subset is designed as a harder negative set: these responses contain common reflection markers but do not revise or overturn any previous conclusion. Because naturally occurring examples of this type are difficult to collect with controlled quality, we synthesize

| Validation set | # Ex. | Run 1 | Run 2 | Run 3 |
|---|---|---|---|---|
| MATH500-SR (genuine) | 25 | 25 | 25 | 24 |
| FineWeb-Edu (no refl.) | 25 | 25 | 25 | 25 |
| Synthetic (surface) | 25 | 25 | 25 | 25 |

Table 7: Judge validation over three repeated runs. Entries report the number of correct labels.

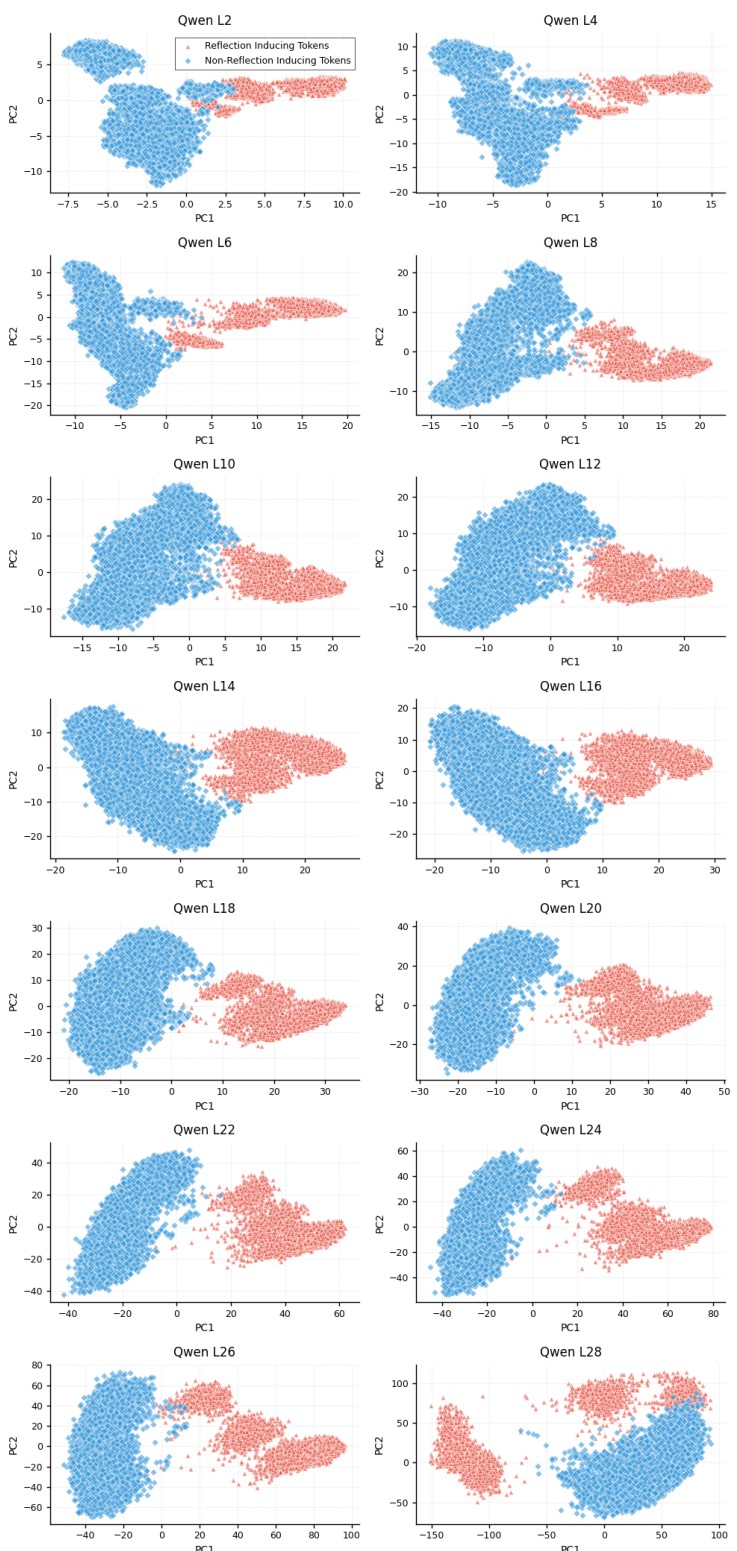

Figure 4: PCA visualization of hidden states from the Qwen2.5-1.5B model across 14 layers.

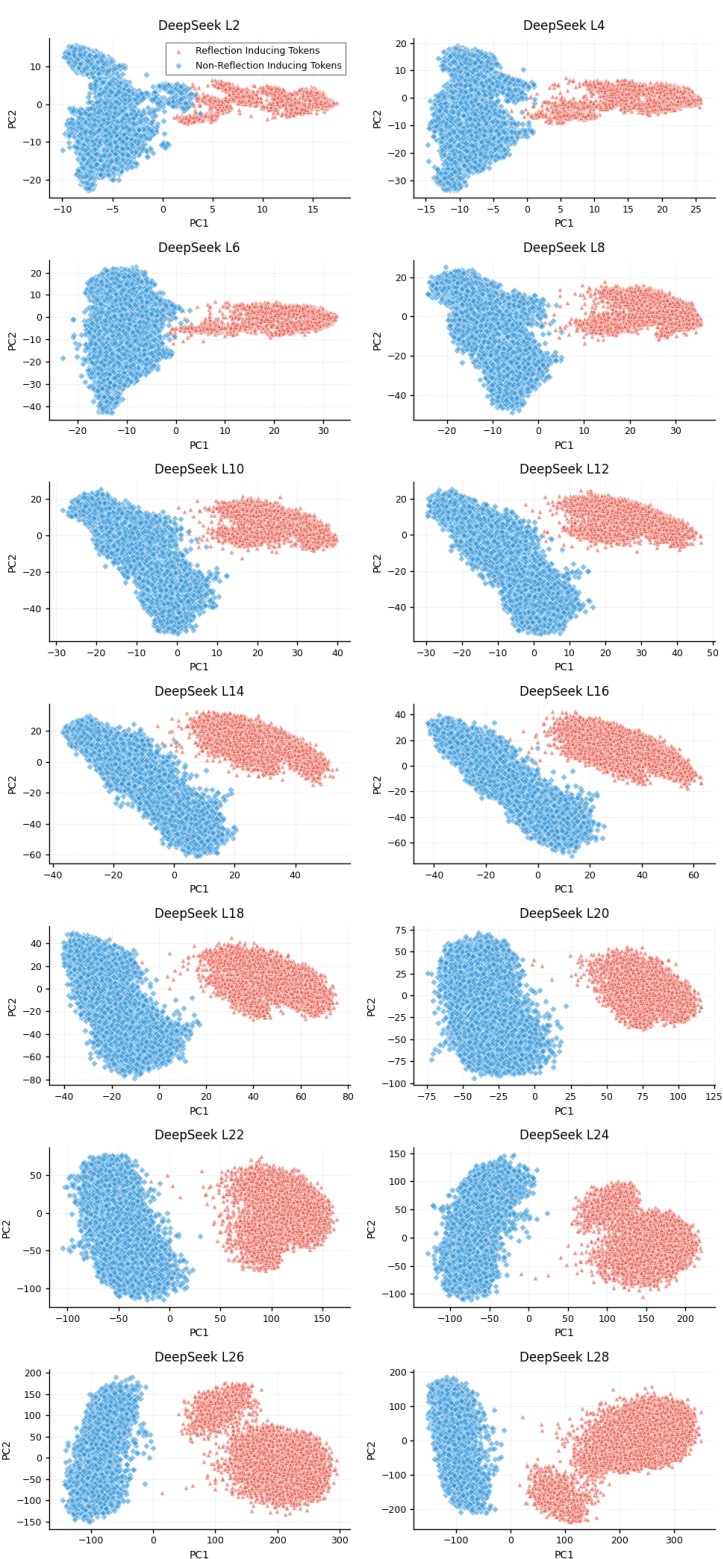

Figure 5: PCA visualization of hidden states from the DeepSeek-R1-Distill-Qwen-1.5B model across 14 layers.

25 hard-negative reasoning trajectories that are rich in reflection-related language but contain no genuine reconsideration.

We use `Gemini-3-Flash` as the judge with its default decoding configuration. For each example, the judge receives the full question and the full model response and is instructed to return a JSON object containing one of three labels: `NO_REFLECTION`, `SURFACE_REFLECTION`, or `GENUINE_SELF_REFLECTION`. We evaluate each example in three independent runs using the same prompt and decoding configuration. The results are reported in Table 7.

For vector construction and reflection-rate analysis, we convert the three-way classification into a binary label by treating `GENUINE_SELF_REFLECTION` as reflective and merging `NO_REFLECTION` and `SURFACE_REFLECTION` into the non-reflective category. Across the three runs, the judge achieves near-perfect agreement with the manually assigned labels, with only one genuine-reflection example misclassified as surface reflection. The perfect performance on the synthetic surface-reflection subset further suggests that the judge is not relying solely on lexical reflection markers, but can distinguish genuine revision from superficial reflective phrasing.

**Prompt for Synthesizing Surface-Reflection Hard Negatives**   To construct the surface-reflection validation set, we use `GPT-5.5` to solve each of the first 25 MATH500 problems while embedding 4–6 varied reflection markers in the trajectory. The full generation prompt is shown below.

---

**Prompt for generating hard-negative (surface-reflection) trajectories**

```
TASK: Generate 5 "hard-negative" self-reflection chain-of-thought trajectories for a research
dataset.  You are simulating the CoT output of a reasoning LLM solving math problems.  These
trajectories are HARD NEGATIVES for a self-reflection judge:  they must LOOK reflective
(contain hesitation markers) but NOT actually be reflective (no real revision).

## INPUT
Read FILE_PATH.

## WHAT TO PRODUCE (per problem)
A chain-of-thought reasoning trajectory (~2000 tokens, roughly 1400-1800 words of
English+math) that:

1.  **Genuinely attempts to SOLVE the math problem.** Work through it step by step.  Arrive at
the ground-truth answer (given in the input's "answer" field) where feasible.  Wrap the final
answer in \boxed{}.

2.  **Contains SEVERAL (4-6) reflection markers NATURALLY embedded in flowing deliberative
prose.** Use varied markers:  "wait", "let me think about this", "hmm", "hold on", "let
me reconsider", "actually", "let me make sure", "let me double-check", "but wait", "let me
verify".  Spread them throughout (not clustered), embedded organically in long meandering
reasoning.

3.  **CRITICAL -- AFTER EACH MARKER, the trajectory must NOT do any of these (doing any =
FAILURE):**
- Identify a specific error in prior reasoning (e.g.  "wait, I made a sign error in step 3")
- Correct a previously computed value (e.g.  "wait, 3×4 isn't 11, it's 12, so the answer
changes to...")
- Backtrack and change the solution direction based on finding a mistake
- Re-derive a result using a different method and get a different/corrected value

**INSTEAD, after each marker, the reasoning must:**
- Restate or rephrase what was already established ("wait, let me make sure -- yes, we have
x^2 + 2x - 3 = 0, that's right, the setup is correct")
- Deliberate about the NEXT step without revising the past ("hmm, now how should I factor this
quadratic...")
```

```
- Express caution/verification that CONFIRMS the prior step ("let me double check:  3² = 9, plus
2·3 = 6, minus 3 = 12...  yes matches, moving on")
- Ponder approach choice WITHOUT changing an already-stated result

**The markers are PERFORMATIVE (surface-level hesitation/discourse), but NO prior conclusion is
ever overturned or corrected.**

## STYLE
- Write like a real long-form reasoning model (DeepSeek-R1 / Qwen-CoT style):  verbose,
exploratory, sometimes repetitive, with natural transitions.
- Vary the markers -- do NOT use "wait" every time.  Mix in "hmm", "let me think", "actually,
...", "hold on", "let me reconsider the setup", "let me make sure I haven't missed anything".
- The text should be LONG, deliberative, meandering -- NOT terse step lists.  Include
motivation, setup, intermediate confirming checks (that confirm, not correct), and forward
progression.
- Do NOT use meta-commentary like "I am now reflecting" or "as a reasoning model".  Stay
in-character.
- Do NOT write the trajectory as a list of bullet points.  Write flowing paragraphs of
reasoning.

## WHAT NOT TO DO
- Do NOT mention labels "SURFACE_REFLECTION", "GENUINE_SELF_REFLECTION", "NO_REFLECTION".
- Do NOT mention "judge", "validation set", "hard negative", "reviewer", "fake reflection".
- The trajectory must read as a genuine model CoT, not as a constructed example.

## OUTPUT FORMAT
For each of your 5 problems, write a JSON file to:
FILE_PATH

JSON schema:
{
  "problem_id":  <int>,
  "unique_id":  "<from input>",
  "problem":  "<full problem text from input>",
  "ground_truth_answer":  "<from input>",
  "subject":  "<from input>",
  "level":  <int>,
  "trajectory":  "<your ~2000 token CoT here, as a single string with \n for line breaks>",
  "generator_model":  "GPT 5.5"
}

## VERIFICATION (mandatory self-check per trajectory)
- [ ] Length is ~2000 tokens (1400-1800 words).
- [ ] 4-6 reflection markers, naturally embedded, varied.
- [ ] NO genuine error-correction anywhere.  After every marker, the prior conclusion is
CONFIRMED or the deliberation moves FORWARD.
- [ ] Math is coherent; arrives at the ground-truth answer (wrapped in \boxed{}).
- [ ] No leaked judge terminology.
- [ ] JSON is valid (parseable).
```

**Judge prompt.**   The full prompt used for identifying self-reflection instances is shown below.

## Prompt for identifying self-reflection instances

```
You are evaluating whether a model response exhibits genuine self-reflection.
```

```
Definition:
Genuine self-reflection means that the model explicitly reconsiders, questions, checks,
backtracks on, or revises its earlier reasoning.  This must go beyond superficial discourse
markers such as "wait", "hold on", or "let me think".  A response counts as genuine
self-reflection only if there is evidence that the model revisits prior reasoning and
meaningfully changes, re-evaluates, or re-derives it.

Label the response as one of:
- NO_REFLECTION: no reflective behavior
- SURFACE_REFLECTION: contains reflection-like phrases, but no substantive reconsideration of
prior reasoning
- GENUINE_SELF_REFLECTION: explicitly revisits prior reasoning and meaningfully re-evaluates,
revises, or corrects it

Instructions:
1.  Read the full question and full response.
2.  Focus on behavior, not keywords alone.
3.  Do not classify as genuine self-reflection unless the response refers back to earlier
reasoning and performs a meaningful check, correction, backtrack, or revision.
4.  Prefer precision over recall.
5.  If the response is labeled as GENUINE_SELF_REFLECTION, identify all reflection occurrences
in the response, not just the first one.
6.  For each reflection occurrence, return the reflection phrase, the corresponding reflection
token, and a short evidence quote.
7.  If there is no genuine self-reflection, return an empty list for "reflection_instances".

Return JSON only:
{
"label":  "NO_REFLECTION | SURFACE_REFLECTION | GENUINE_SELF_REFLECTION",
"confidence":  0.0--1.0,
"reflection_instances":  [
{
"reflection_phrase":  "...",
"reflection_token":  "...",
"evidence_quote":  "..."
}
]
}

Question:
{question}

Response:
{response}
```

**Qualitative examples.**   To illustrate the behavioral boundary that the judge must draw, we contrast the two reflection categories on the same MATH500 problem. The first box shows a synthesized surface-reflection trajectory: it contains even more reflection markers, yet every marker is followed only by confirmation of an already-established result or deliberation about the next step, and no prior conclusion is ever revised. The second box shows a genuine self-reflection example generated by Qwen2.5-7B: after each highlighted marker, the model actively re-examines its earlier conclusion, verifying it by back-conversion and, crucially, testing and rejecting the competing angle $\frac{3\pi}{2}$. The two traces are nearly indistinguishable at the marker level, which is precisely what makes the synthesized set a meaningful test.

**Hard-negative (surface-reflection) trajectory**

**Problem.** Convert the point $(0, 3)$ in rectangular coordinates to polar coordinates. Enter your answer in the form $(r, \theta)$, where $r > 0$ and $0 \leq \theta < 2\pi$.
**Ground-truth answer.** $\left(3, \frac{\pi}{2}\right)$

---

We are asked to convert the point $(0, 3)$ from rectangular (Cartesian) coordinates into polar coordinates, expressing the answer in the form $(r, \theta)$ with the conditions that $r > 0$ and $0 \leq \theta < 2\pi$. Before I touch any formula, I want to build some intuition for what these two coordinate systems are doing, because the condition that the $x$-coordinate is exactly zero is going to matter when I reach the angle, and I want to be deliberate about how I handle it rather than blindly substituting into a formula that could quietly mislead me.

In the rectangular system, a point is pinned down by two signed displacements: how far left or right of the origin it sits (that is $x$), and how far up or down it sits (that is $y$). In the polar system, the same point is pinned down differently: we give its straight-line distance from the origin, called $r$, together with the angle $\theta$ that the ray from the origin through the point makes with the positive $x$-axis, measured counterclockwise. These two descriptions are just different address systems for the same plane, and they are linked by the relations $x = r\cos\theta$ and $y = r\sin\theta$ in one direction, and $r = \sqrt{x^2 + y^2}$ with $\tan\theta = y/x$ (when $x \neq 0$) in the other direction.

The given point is $(x, y) = (0, 3)$. I picture it sitting three units directly above the origin, plumb on the vertical axis, with no horizontal offset at all. That picture alone already suggests that the ray from the origin to the point points straight up, which is the direction we associate with ninety degrees, but I will confirm this rigorously through the formulas rather than trusting the picture by itself.

I begin with the radius, which is the more straightforward of the two quantities. Using the distance formula, $r = \sqrt{x^2 + y^2} = \sqrt{0^2 + 3^2} = \sqrt{0 + 9} = \sqrt{9} = 3$. Because the problem insists on $r > 0$, I keep the positive root, which is the natural choice here regardless. So the radial coordinate is settled: $r = 3$. This also matches the bare geometric fact that the point is three units away from the origin, since the segment from $(0, 0)$ to $(0, 3)$ is vertical with length exactly 3. The formula and the picture agree, which is a good consistency check right at the start. There is a slightly different way to think about the radius that reinforces the result through a second line of reasoning: the radius measures the hypotenuse of the right triangle whose legs are the horizontal displacement $|x| = 0$ and the vertical displacement $|y| = 3$, and by the Pythagorean theorem $r^2 = 0^2 + 3^2 = 9$, giving $r = 3$ once we take the positive root. This right-triangle picture is exactly where the formula $r = \sqrt{x^2 + y^2}$ comes from in the first place, so reasoning through the triangle is not really a separate computation but the conceptual origin of the formula, and it confirms that $r = 3$ is a geometric necessity rather than just a formulaic output.

Now I turn to the angle $\theta$, and here I have to be careful. The tangent-based formula gives $\tan\theta = y/x = 3/0$, which is undefined, so the arctangent route is not directly available to me. **Hmm,** that is not really a problem but rather a signal: division by zero here is telling me that the point lies on the vertical axis, where the usual arctangent formula is the wrong tool. I will reason about the angle geometrically and then cross-check it algebraically through the reverse conversion formulas, which do not suffer from any division-by-zero issue.

The point $(0, 3)$ lies on the positive $y$-axis. By definition, the positive $y$-axis is the ray obtained by rotating the positive $x$-axis counterclockwise through a right angle, which is $\pi/2$ radians. Therefore the ray from the origin to $(0, 3)$ makes an angle of $\pi/2$ with the positive $x$-axis. So my candidate angle is $\theta = \pi/2$. **Let me make sure** this lies in the required range: the problem asks for $0 \leq \theta < 2\pi$, and $\pi/2$ is certainly within that interval (it is roughly 1.57, well between 0 and about 6.28). So $\pi/2$ is a valid principal value, and there is no range violation to worry about.

Now I confirm both coordinates simultaneously by converting back from polar to rectangular using $x = r\cos\theta$ and $y = r\sin\theta$. With $r = 3$ and $\theta = \pi/2$, I get $x = 3\cos(\pi/2)$. The value $\cos(\pi/2)$ is 0, so $x = 3 \cdot 0 = 0$. Good, that recovers the given $x = 0$. Then $y = 3\sin(\pi/2)$. The value $\sin(\pi/2)$ is 1, so $y = 3 \cdot 1 = 3$. That recovers the given $y = 3$ exactly. The round trip through the formulas reproduces the original point, which tells me that the pair $(r, \theta) = (3, \pi/2)$ is correct.

**Let me double-check** the underlying trigonometric values themselves, so that the back-conversion is resting on solid ground rather than on a misremembered table. At angle $\pi/2$, the corresponding point on the unit circle is the topmost point, with coordinates $(\cos(\pi/2), \sin(\pi/2)) = (0, 1)$. Reading off those coordinates gives $\cos(\pi/2) = 0$ and $\sin(\pi/2) = 1$, which are exactly the values I used. So the back-conversion is sound.

**Let me reconsider** the setup from the standpoint of uniqueness, because the conditions $r > 0$ and $0 \leq \theta < 2\pi$ are there precisely to guarantee a unique polar address, and I want to be sure my answer is the only one that qualifies. With $r > 0$ enforced, I cannot play the negative-radius trick of writing the same point as $(r, \theta) = (-3, 3\pi/2)$, because that would violate $r > 0$. And with $\theta$ restricted to a half-open interval of length $2\pi$, each direction from the origin is hit by exactly one angle, so the direction straight up corresponds to the single value $\theta = \pi/2$ in the allowed range. Hence the representation $(3, \pi/2)$ is unique under the stated constraints, and I do not need to worry about competing answers.

I also **cross-check** the angle using the two-argument arctangent function $\operatorname{atan2}(y, x)$, which is designed to handle points on the axes gracefully by incorporating the signs of both coordinates. For $(x, y) = (0, 3)$, $\operatorname{atan2}(3, 0)$ returns $\pi/2$, because the point is on the positive $y$-axis. This agrees with both my geometric reasoning and my back-conversion, and it gives me a third independent route to the same angle. The agreement across three different methods, geometric reasoning, the reverse conversion formulas, and the two-argument arctangent, makes me confident that $\theta = \pi/2$ is correct.

Reflecting briefly on why the direct formula $\arctan(y/x)$ failed here: it simply was not designed for points where $x = 0$, because the slope $y/x$ becomes infinite and the arctangent cannot distinguish a vertical-up direction from a vertical-down direction without sign information. The two-argument version resolves exactly this ambiguity, and the geometric picture sidesteps it entirely. So the failure of the naive formula was expected and is not a sign of any mistake on my part; it just means I had to choose a better-suited method, which I did.

Bringing everything together: the radius is $r = 3$, confirmed both by the distance formula and by direct geometric measurement; the angle is $\theta = \pi/2$, confirmed by geometric reasoning, by the reverse conversion formulas, by the standard trigonometric values on the unit circle, by the uniqueness argument under the given range constraints, and by the two-argument arctangent. Both quantities satisfy the required conditions $r > 0$ and $0 \leq \theta < 2\pi$, and the back-conversion reproduces the original point $(0, 3)$ exactly. The polar coordinates are therefore $(3, \pi/2)$.

The answer is $\boxed{\left(3, \dfrac{\pi}{2}\right)}$.

---

### Self-reflection example from Qwen2.5-7B

We need to convert the rectangular coordinate point $(0, 3)$ into polar coordinates. In rectangular coordinates, a point is written as $(x, y)$, while in polar coordinates, the same point is written as $(r, \theta)$, where $r$ is the distance from the origin and $\theta$ is the angle measured counterclockwise from the positive $x$-axis. The problem also tells us that the answer should satisfy $r > 0$ and $0 \leq \theta < 2\pi$, so we need to choose the standard polar representation with a positive radius and an angle in this interval.

For the point $(0, 3)$, we have

$$x = 0, \qquad y = 3.$$

The radius $r$ is computed using

$$r = \sqrt{x^2 + y^2}.$$

Substituting $x = 0$ and $y = 3$, we get

$$r = \sqrt{0^2 + 3^2} = \sqrt{9} = 3.$$

So the radius is 3. This makes sense geometrically because the point $(0, 3)$ is exactly 3 units above the origin.

Now we need to find the angle $\theta$. Usually, we can use

$$\tan \theta = \frac{y}{x}.$$

However, in this case $x = 0$, so $\frac{y}{x}$ is undefined. That means we should not blindly use the arctangent formula. Instead, we should think about the location of the point geometrically.

The point $(0, 3)$ has $x = 0$ and $y > 0$, so it lies on the positive $y$-axis. The angle from the positive $x$-axis to the positive $y$-axis is

$$\frac{\pi}{2}.$$

Therefore, the angle should be

$$\theta = \frac{\pi}{2}.$$

**Let me check that carefully.** If the polar coordinates are

$$\left(3, \frac{\pi}{2}\right),$$

then converting back to rectangular coordinates should give the original point. The conversion formulas are

$$x = r\cos\theta, \qquad y = r\sin\theta.$$

Using $r = 3$ and $\theta = \frac{\pi}{2}$, we get

$$x = 3\cos\frac{\pi}{2} = 3 \cdot 0 = 0,$$

and

$$y = 3\sin\frac{\pi}{2} = 3 \cdot 1 = 3.$$

So the rectangular coordinates become

$$(0, 3),$$

which exactly matches the original point.

Let me also make sure there is no issue with the required angle range. The problem requires

$$0 \le \theta < 2\pi.$$

The angle $\frac{\pi}{2}$ is in this interval, since it is greater than 0 and less than $2\pi$. Also, the radius $r = 3$ is positive, so it satisfies $r > 0$.

**Wait**, another possible angle that gives an $x$-coordinate of 0 is $\frac{3\pi}{2}$, but that angle points downward along the negative $y$-axis. If we used

$$\theta = \frac{3\pi}{2},$$

then

$$x = 3\cos\frac{3\pi}{2} = 0,$$

but

$$y = 3\sin\frac{3\pi}{2} = -3.$$

That would give the point $(0, -3)$, not $(0, 3)$. So $\frac{3\pi}{2}$ is not correct for this point.

Thus, the correct angle is $\frac{\pi}{2}$, and the correct radius is 3. Therefore, the polar coordinates of the point $(0, 3)$ are

$$\boxed{\left(3, \frac{\pi}{2}\right)}.$$

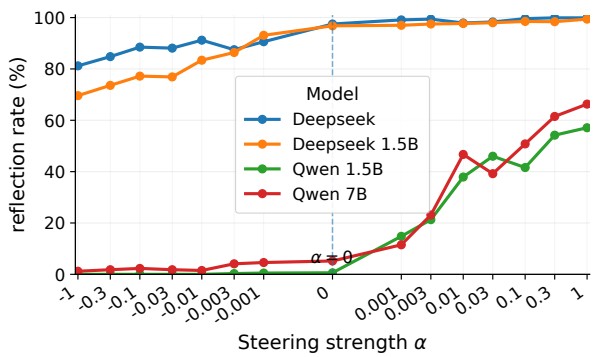

(a) Reflection rate under different steering strengths.

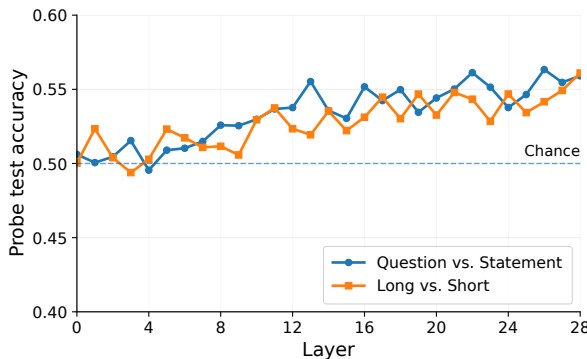

(b) Linear-probe accuracy for control contrasts.

Figure 6: Additional diagnostic analyses for self-reflection steering. **(a)** Reflection rate as a function of steering strength $\alpha$ for both 1.5B and 7B models, including DeepSeek-R1-Distill-Qwen and Qwen2.5 variants. **(b)** Linear-probe test accuracy on DeepSeek-R1-7B for two generic control contrasts, question vs. statement and long vs. short, across layers. The question vs. statement contrast uses 100 examples and probes the final-token activation of each sentence. The long vs. short contrast uses another 100 examples and probes the final-token activation of each response, with long responses defined as generations longer than 4K tokens and short responses as the remaining generations.

## D  Projected Steering as State-Dependent Adaptive Control

We further discuss why projected steering can be more effective than a fixed additive intervention. Recall that our intervention is defined as

$$\widehat{\boldsymbol{h}}^{(\ell)} = \boldsymbol{h}^{(\ell)} + \alpha \left\langle \boldsymbol{h}^{(\ell)}, \boldsymbol{v}^{(\ell)} \right\rangle \boldsymbol{v}^{(\ell)} = \boldsymbol{h}^{(\ell)} + \alpha_{\text{eff}}(\boldsymbol{h})\boldsymbol{v}^{(\ell)}, \qquad \alpha_{\text{eff}}(\boldsymbol{h}) = \alpha \left\langle \boldsymbol{h}^{(\ell)}, \boldsymbol{v}^{(\ell)} \right\rangle. \tag{6}$$

Thus, projected steering can be interpreted as a simple state-dependent adaptive intervention: the effective steering strength is not a fixed constant, but varies with the current hidden state.

This differs from additive steering, which applies the same perturbation $\alpha \boldsymbol{v}^{(\ell)}$ at every position regardless of whether the current token is reflection-relevant. In contrast, projected steering increases the intervention when the hidden state already contains a strong component along the reflection direction, and decreases it when this component is weak. In this sense, the scalar projection acts as a lightweight adaptive gain.

## E  Additional Diagnostics

### E.1  Steering Strength and Linear Separability

Figure 6a shows that the steering strength $\alpha$ provides direct control over reflection frequency. For the fine-tuned model, reflection is already frequent at $\alpha = 0$; positive steering further increases it, while negative steering reduces but does not fully remove it. For the pretrained model, reflection is rare without steering, but positive $\alpha$ can substantially elicit it. This supports the view that self-reflection is a latent behavior that can be continuously modulated by the learned direction.

Figure 6b tests whether our linear separability result is driven by simple surface-level confounds. On DeepSeek-R1-7B, probes for question-vs.-statement and long-vs.-short contrasts remain close to chance across layers, unlike the near-perfect self-reflection probe. This suggests that the self-reflection direction is not merely capturing generic syntax or response length, but a more specific latent state associated with reflective revision.

Table 8: Control-direction comparison on MATH500. Random directions are norm-matched to SR and averaged over five independent draws. The verbosity direction is constructed from long-vs.-short responses, where long responses are defined as outputs longer than 4K tokens for DeepSeek-R1-7B and longer than 2K tokens for Qwen2.5-7B. Unlike the main results in Table 1, which use greedy decoding, this table uses stochastic decoding with different random seeds to estimate run-level uncertainty. For all non-random directions, we report the mean and standard deviation over three runs with different seeds.

| Model | Metric | None | Random | Verbosity | Correctness | SR |
|---|---|---|---|---|---|---|
| DeepSeek-R1-7B | PASS@1 | 92.7±1.9 | 80.3±7.4 | 87.8±3.6 | 91.1±2.2 | 93.5±1.2 |
| | Len. | 3585±293 | 7312±1418 | 6124±1206 | 4718±537 | 8959±916 |
| Qwen2.5-7B | PASS@1 | 44.8±1.9 | 36.9±6.8 | 41.7±3.9 | 46.2±2.5 | 56.6±2.7 |
| | Len. | 1294±15 | 2811±392 | 5476±489 | 2093±301 | 2671±462 |

Table 9: Control experiments for reflection-inducing prefixes on MATH500. "Reflective" uses the original reflective CoT prefix. "Rewritten" is produced by prompting another LLM to rewrite the reflective prefix while keeping the final ten tokens unchanged. "Len.-matched" uses a non-reflective CoT prefix with comparable token length. "Style-matched" removes explicit self-reflection-inducing content, including the final ten tokens. "Shuffled" randomly shuffles the reflective prefix.

| Metric | Original | Reflective | Rewritten | Len.-matched | Style-matched | Shuffled |
|---|---|---|---|---|---|---|
| Reflection Rate | 0.4 | 38.9 | 34.2 | 0.1 | 0.0 | 0.0 |
| Acc. | 27.5 | 32.1 | 31.4 | 28.6 | 28.0 | 10.8 |

## E.2 Control Directions

We compare SR steering against several control directions to test whether its effect can be explained by generic activation perturbations, increased verbosity, or correctness-related representations. Specifically, we use norm-matched random directions, a verbosity direction constructed from long-vs.-short responses, and a correctness direction constructed from correct-vs.-incorrect responses. For these control directions, the reported variance comes from constructing directions with different sampled example sets.

As shown in Table 8, norm-matched random directions substantially degrade performance, indicating that arbitrary movement in activation space is not sufficient. The verbosity direction increases generation length but does not yield consistent accuracy gains, suggesting that longer responses alone do not explain the effect of SR steering. The correctness direction gives mixed results and remains weaker than SR, especially on Qwen2.5-7B. Together with the $\alpha$-sweep analysis in Figure 6a, these results suggest that SR steering is not merely a length, correctness, or random-perturbation effect, but instead captures a more specific reflection-related state.

## E.3 Prefix-Control Experiments

We further test whether reflection-inducing prefixes work through coherent reflective content rather than superficial prefix properties. To this end, we compare the original reflective prefix with rewritten, length-matched, style-matched, and shuffled variants.

As shown in Table 9, the reflective prefix substantially increases both judge-labeled reflection rate and accuracy over the vanilla prompt. The rewritten prefix preserves most of this effect, indicating that the result is not tied to the exact wording of the original prefix. In contrast, the length-matched and style-matched controls produce almost no self-reflection and only minor accuracy changes, suggesting that neither prefix length nor generic CoT style is sufficient. The shuffled prefix substantially hurts performance, further indicating that coherent reflection-relevant content, rather than unordered reflection-related tokens, is necessary.

Table 10: Additional generalization and ablation results. **(a)** reports HumanEval pass@1 and average generation length under greedy decoding. **(b)** varies the number of examples used to construct the SR direction on Qwen2.5-7B and evaluates on 50 held-out MATH500 questions.

(a) HumanEval results

| Model | Setting | Pass@1 | Len. |
|---|---|---|---|
| DeepSeek-R1-7B | Base. | 93.5 | 6943 |
| | SR Enh. | 96.7 | 8214 |
| | SR Supp. | 91.4 | 4341 |
| Qwen2.5-7B | Base. | 82.8 | 3493 |
| | SR Enh. | 89.5 | 6114 |

(b) Sample-count ablation

| $N$ | No steer. | 25 | 50 | 100 | 250 | 500 |
|---|---|---|---|---|---|---|
| SR rate | 0.4 | 30.9 | 29.8 | 31.6 | 25.2 | 24.9 |
| Acc. | 44.8 | 52.1 | 57.2 | 56.6 | 54.3 | 51.6 |
| Len. | 1294 | 1973 | 2340 | 2671 | 2226 | 2954 |

### E.4 Generalization to Code Generation

Table 10a evaluates whether SR steering transfers beyond mathematical reasoning. On HumanEval, SR enhancement improves pass@1 for both DeepSeek-R1-7B and Qwen2.5-7B under greedy decoding. For DeepSeek-R1-7B, SR suppression decreases both pass@1 and average generation length, suggesting that suppressing self-reflection can also reduce deliberative behavior in code generation. These results provide initial evidence that SR steering is not restricted to math-style reasoning, though broader non-math evaluations remain an important direction for future work.

### E.5 Sample-Count Ablation

Table 10b studies how many construction examples are needed to obtain an effective SR direction. On Qwen2.5-7B, the steering effect is already visible with $N = 25$ examples and is strongest around $N = 50$–100. Using more construction examples does not monotonically improve accuracy or reflection rate. This suggests that the method does not require a very large labeled set, while also indicating that direction quality depends on more than sample count alone.

### E.6 AIME 2024 Uncertainty Analysis

Because AIME 2024 is a small evaluation set, we additionally report uncertainty estimates over repeated evaluation runs and item-level paired bootstrap intervals. This analysis is intended to distinguish improvements that are consistent across items from changes that may be explained by evaluation variance.

As shown in Table 11, BF does not provide a reliable improvement over Vanilla. In contrast, SR Enhanced yields a positive gain on DeepSeek-7B and a substantially larger gain on Qwen2.5-7B. These results support the conclusion that the SR gains on AIME 2024 are not solely due to run-level variation, while also showing that the magnitude of the improvement is model-dependent.

At the item level, the additional problems solved by SR Enhanced relative to BF are concentrated on a small subset of cases:

- Run 1: Problem 14;
- Run 2: Problems 12 and 14;
- Run 3: Problem 12;
- Run 4: none.

This pattern suggests that SR Enhanced mainly helps near-miss, self-check-correctable problems, where additional verification can refine an already close solution. For more difficult AIME problems that remain beyond the capability of the model, simply enhancing self-reflection is not expected to solve them.

Taken together, the right-skewed interval for DeepSeek-R1-7B suggests that the outcome-level gain should not be interpreted as a uniform lift over all AIME problems. Our interpretation is that SR Enhanced mainly helps a subset of near-miss, self-check-correctable cases, where additional verification can refine an already

Table 11: AIME 2024 uncertainty analysis over four repeated evaluation runs. Unlike the main results in Table 1, which use greedy decoding, this table uses stochastic decoding with different random seeds to estimate run-level uncertainty. For steering-based methods, the random seed also affects the sampled subset of self-reflection pairs used to construct the steering vector. For each method, we report the per-run number of correctly solved problems out of 30, mean Pass@1 with standard deviation across runs, and average generation length. For each method, $\Delta$ denotes the difference relative to Vanilla, computed using the average accuracy over the four runs. The 95% confidence intervals are computed by paired bootstrap resampling over evaluation items, where each item is represented by its average accuracy over the four runs.

| Model | Method | Runs | Pass@1 | Avg. Len. | Mean $\Delta$ [95% CI] |
|---|---|---|---|---|---|
| DeepSeek-R1-7B | Vanilla | 16, 16, 17, 18 | 55.8±3.1 | 4737±494 | – |
| | BF | 15, 16, 17, 18 | 54.5±4.3 | 9844±1656 | −1.3 [-3.2, +2.4]] |
| | SR Enhanced | 16, 18, 18, 18 | 58.2±2.8 | 6352±1249 | +2.4 [+0.7, +6.9] |
| | SR Suppressed | 15, 15, 16, 18 | 52.9±4.7 | 3692±685 | −2.9 [-3.6, +0.5] |
| Qwen2.5-7B | Vanilla | 1, 1, 1, 2 | 3.9±1.7 | 2285±297 | – |
| | BF | 1, 2, 2, 2 | 5.3±1.7 | 4526±401 | +1.4 [-1.0, +3.7] |
| | SR Enhanced | **4, 5, 5, 6** | **16.1±2.7** | 2941±486 | +12.2 [+10.8, +20.4] |

Table 12: Efficiency analysis on two representative 7B models. Latency denotes average end-to-end wall-clock generation time per example. Throughput is measured as generated tokens per second. Peak memory is the maximum allocated GPU memory during generation, and $\Delta$ Mem. is measured relative to the vanilla setting for the same model. All measurements use the same hardware, batch size, maximum generation length, and greedy decoding.

| Model | Setting | Avg. Len. | Latency (s) | Throughput (tok/s) | Peak Mem. (GB) | $\Delta$ Mem. |
|---|---|---|---|---|---|---|
| DeepSeek-R1-7B | Vanilla | 3585 | 69.1 | 51.9 | 20.4 | – |
| | BF | 7111 | 141.4 | 50.3 | 21.1 | +0.7 |
| | SR Enhanced | 8959 | 178.5 | 50.2 | 21.5 | +1.1 |
| Qwen2.5-7B | Vanilla | 1294 | 24.6 | 52.6 | 19.8 | – |
| | BF | 3986 | 78.6 | 50.7 | 20.5 | +0.7 |
| | SR Enhanced | 2671 | 52.9 | 50.5 | 20.2 | +0.4 |

close solution. SR Enhanced may also alter the reasoning trajectory on examples whose final correctness does not change, but such trajectory-level changes are distinct from the measured accuracy gain.

### E.7 Efficiency Overhead

We also measure the practical overhead of SR steering in terms of wall-clock latency, decoding throughput, and peak GPU memory. All measurements use the same hardware, batch size, maximum generation length, and greedy decoding setup.

As shown in Table 12, SR steering increases end-to-end latency mainly because it changes the number of generated tokens. The per-token decoding throughput remains similar to the vanilla model, and the peak-memory overhead is small. This is expected because the intervention adds a fixed activation-space direction at the selected layer, without modifying model parameters or storing additional KV-cache states. Thus, the main computational cost of SR steering comes from induced longer generations, rather than from the activation intervention itself.

### E.8 Correctness-Conditioned Reasoning-Trace Quality

A potential concern with the aggregate ROSCOE-SS results in Table 2 is that steering could degrade reasoning traces in a way that is masked by averaging: for example, if SR suppression pushed the model toward locally coherent but less reliable shortcut reasoning, the resulting traces might retain similar semantic-coherence scores while being meaningfully worse. To partially address this, we condition the analysis on

| Setting | ROSCOE-SS$_{\text{incorrect}}$ | ROSCOE-SS$_{\text{correct}}$ |
|---|---|---|
| Vanilla | 0.557 | 0.579 |
| SR Enhanced | 0.551 | 0.573 |
| SR Suppressed | 0.562 | 0.584 |

Table 13: Correctness-conditioned ROSCOE-SS on MATH500 for DeepSeek-R1-1.5B. We report ROSCOE-SS separately for examples with correct and incorrect final answers under each steering setting.

final-answer correctness, computing ROSCOE-SS separately for examples whose final answers are correct and incorrect on MATH500 with DeepSeek-R1-1.5B.

As shown in Table 13, ROSCOE-SS remains close to Vanilla within both the correct and incorrect groups, suggesting that the aggregate results in Table 2 do not mask degradation on failure cases.

## F    Broader Impact

This work provides an interpretable way to modulate self-reflection at inference time using an activation direction and a scalar coefficient. This may help allocate reasoning effort more selectively, improving efficiency on simple queries while preserving the option to increase deliberation on harder or higher-risk tasks.

However, reflection control also has dual-use risks. Suppressing reflection may reduce a model's tendency to verify intermediate steps, reconsider mistakes, or perform informal safety checks, which is concerning in domains such as medicine, law, and finance. Conversely, enhancing reflection may produce longer and more convincing rationalizations without necessarily improving correctness. Because activation-level steering changes the model's internal tendency to reflect in a way that may not be visible to users, it may also affect how users perceive model reliability. Therefore, reflection steering should be used conditionally and evaluated not only for accuracy and length, but also for calibration, robustness, and safety behavior.

