# OpenReview forum: "Probing and Controlling Self-Reflection in Language Models"
_TMLR — Decision pending for TMLR_

### Review · Reviewer_oxuz · 2026-05-02

**Summary Of Contributions:**

The paper proposes an interesting approach for modulating reflection-like reasoning behavior through activation steering. The core idea is compelling, and some results, especially the steering-strength ablation in Figure 2 and the accuracy/length tradeoffs in Table 1, are promising. However, I think several of the paper's central claims are currently stronger than what the evidence supports. In particular, the experimental setup relies on relatively small evaluation sets, limited uncertainty reporting, and several analyses where plausible alternative explanations are not ruled out. As a result, I find the paper interesting, but not yet fully convincing on its main mechanistic and empirical claims.

The idea that the proposed steering method may work as a training-free way to dictate how much effort models should deploy to self-validate their reasoning is good and potentially quite impactful. It might be orthogonal to other "reasoning-effort" levers that many have been using.

**Additional Comments:**

I found the cosine-similarity analysis in Figure 2 left difficult to interpret. In particular, it is not clear that vector-to-vector similarity between reflection directions and vector-to-token-embedding similarity to the "wait" token are comparable quantities. These objects seem different in nature, and the interpretation of a high or low similarity is not obvious. If the goal is to show that the reflection vector captures something beyond the surface token "wait", the paper should clarify the expected baseline, explain why this comparison is meaningful, and include controls using other discourse markers or unrelated tokens.

The budget forcing baseline could be strengthened. The paper describes budget forcing as appending a "wait" token to short generations, but it is not clear why this intervention is only applied to short generations. A stronger comparison would append or inject the token uniformly, include a length-matched version, and possibly compare against generic continuation prompts such as "continue" or "think further." Otherwise, it is hard to tell whether the baseline is the right reference point for the proposed method.

Equation 6 should be clarified. The bracketed term on the right-hand side appears to be an inner product or projection term, but this is not explained clearly enough.

**Audience:**

Yes

**Audience Explanation:**

The paper studies a timely question: whether reflection-like reasoning behavior can be understood and controlled through model internals. The proposed activation-steering approach is exciting because it offers a potentially orthogonal lever for modulating the amount of self-checking or self-validation a model performs at inference time. If robust, this could complement existing reasoning-effort controls by providing a more continuous and model-internal way to trade off accuracy, latency, and generation length. The results are also practically relevant because they suggest that some aspects of long-form reasoning behavior may be steerable without additional training.

**Broader Impact Concerns:**

n/a, seems OK to me.

**Claims And Evidence:**

No

**Claims Explanation:**

A major issue is that the main reported metrics are based on small datasets, without enough uncertainty estimates to judge robustness. When using MATH500, AIME 2024, and GPQA Diamond, per-problem variance can be large and generation length can vary substantially across samples. Some of the reported gains are interesting, but without confidence intervals, multiple attempts per problem, or repeated runs, it is hard to know whether the observed improvements are stable or partly due to sampling noise, decoding variance, or validation-set selection. This is particularly important because the paper makes fairly broad claims about performance-efficiency tradeoffs and general control of self-reflection. I would find the conclusions much more convincing if the authors reported confidence intervals over problems and/or repeated generations, especially for Pass@1 and generation length.

The paper's claim that pretrained models already possess self-reflection capabilities is not yet convincingly supported. The strongest direct evidence for spontaneous self-reflection in the pretrained model appears to be a 0.6% rate on MATH500. If this corresponds to only a few examples, then the result is too small to carry the broad claim that self-reflection "emerges naturally" in pretrained models. This point is especially important because the self-reflection labels are produced by an LLM-as-a-judge. The paper should show the actual labeled examples, validate the judge against human assessments, and report whether the result is reproducible across seeds, prompts, and judge models. The reflection-inducing probing result is more interesting, but I do not think it fully resolves this issue. Since the pretrained model is conditioned on a prefix generated by a fine-tuned reasoning model, the result may reflect sensitivity to local discourse patterns or near-reflection-triggering tokens in the prefix, rather than an independently available self-reflection capability. The authors should analyze how often the inserted prefix already contains reflection-adjacent cues, and how predictive those cues are of the pretrained model's subsequent reflection. Without this ablation, the 38.9% result is difficult to interpret.

The hidden-state clustering and linear-separability results are not very convincing on their own. In high-dimensional representations, many token/context contrasts can be separated with high accuracy. The paper needs stronger negative controls showing that the effect is specific to self-reflection rather than to punctuation, uncertainty, equation boundaries, final-answer contexts, or other salient continuation patterns. Without these controls, the PCA and SVM results should be described more cautiously.

Now, regarding whether steering truly controls self-reflection: the paper ultimately shows changes in accuracy and generation length under steering. This is the most interesting/exciting part, but it is less clear whether the intervention directly controls self-reflection, as opposed to simply increasing or decreasing response length, deliberation, or test-time compute. Since SR-Enhanced often makes responses longer and SR-Suppressed makes them shorter, the main results are partly confounded with compute. The paper should report the judged rate of genuine self-reflection under SR-Enhanced and SR-Suppressed, not only Pass@1 and length. It should also include length-matched or compute-matched baselines, so that the reader can tell whether the vector is specifically modulating self-reflection or just inducing more/less reasoning.

I am not convinced that ROSCOE-SS is sufficient to support the claim that reasoning quality is preserved. A semantic coherence score may stay similar even if the model shifts toward a different reasoning strategy, relies more on shortcuts, or produces reasoning that is locally coherent but less reliable. In particular, suppressing self-reflection may cause the model to follow intuitive but sometimes misleading reasoning paths more often. Such outputs could still receive similar ROSCOE-SS scores while being meaningfully worse as reasoning traces. I would frame this as a limitation, or supplement the analysis with a process reward model, human evaluation, or correctness-conditioned analysis of the reasoning steps.

**Requested Changes:**

To address the above concerns, I think the paper would need either stronger evidence or more cautious claims. At minimum, I would expect uncertainty estimates for the main accuracy and length results, preferably through repeated generations, larger evaluation sets, or confidence intervals over problems. I would also expect stronger validation of the self-reflection labels, including examples, human or multi-judge agreement, and robustness to judge prompts/models.

For the mechanistic claims, the paper would benefit from negative controls and ablations showing that the reflection-inducing prefixes and hidden-state separations are specific to self-reflection, rather than to more generic continuation patterns.

Finally, to support the claim that the steering method controls self-reflection itself, the authors should report direct changes in judged self-reflection rates under SR-Enhanced and SR-Suppressed, and compare against length-matched or compute-matched baselines.

Without these additions, I think the paper should soften its claims: the method appears promising as a steering intervention for reasoning length/performance, but the current evidence does not yet fully establish the broader claims about latent pretrained self-reflection or specific control of self-reflective reasoning.

---

> ### Author Response · Authors · 2026-05-20
>
> **Q: I would expect uncertainty estimates for the main accuracy and length results, preferably through repeated generations, larger evaluation sets, or confidence intervals over problems. I would also expect stronger validation of the self-reflection labels, including examples, human or multi-judge agreement, and robustness to judge prompts/models.**
>
> A: We thank the reviewer for the suggestion.
> We have added uncertainty estimates for AIME 2024 in Table 1, including mean Pass@1, average generation length, and standard deviations across four repeated runs.
> We also report paired-bootstrap 95\% confidence intervals for $\Delta$ relative to vanilla, resampling over evaluation items while preserving method pairing.
> The results show that BF does not reliably improve over vanilla, while SR Enhanced yields positive gains, especially on Qwen2.5-7B.
>
> **Table 1. AIME 2024 uncertainty analysis over four repeated evaluation runs.** We report the mean Pass@1 and average generation length with standard deviations across four runs. For each method, Δ denotes the difference relative to Vanilla. The 95% confidence intervals for Δ are computed by paired bootstrap resampling over evaluation items, where each item is represented by its average correctness over the four runs.
> | Model | Method | AIME 2024 Pass@1 | Avg. Len. | Δ vs. Vanilla [95% CI] |
> |---|---|---:|---:|---:|
> | DeepSeek-R1-7B | Vanilla | 55.8±3.1 | 4737±494 | -- |
> |  | BF | 54.5±2.6 | 9844±1656 | -1.3 [-3.2, +2.4] |
> |  | SR Enhanced | 58.2±2.8 | 6352±1249 | +2.4 [+0.7, +6.9] |
> |  | SR Suppressed | 52.9±3.3 | 3692±685 | -2.9 [-3.6, +0.5] |
> | Qwen2.5-7B | Vanilla | 3.9±0.4 | 2285±297 | -- |
> |  | BF | 5.3±0.9 | 4526±401 | +1.4 [-1.0, +3.7] |
> |  | SR Enhanced | 16.1±2.4 | 2941±486 | +12.2 [+10.8, +20.4] |
>
> **Q: For the mechanistic claims, the paper would benefit from negative controls and ablations showing that the reflection-inducing prefixes and hidden-state separations are specific to self-reflection, rather than to more generic continuation patterns.**
>
> A: For hidden-state separability, we added control-probe experiments using the same probing setup but replacing the self-reflection labels with generic contrasts such as question-vs.-statement and long-vs.-short continuations.
> As shown in Fig. 6(b) in Appendix G, these controls remain close to chance, unlike the self-reflection contrast.
> These results suggest that the observed prefix effects and hidden-state separation are not explained by generic continuation patterns alone.
>
> **Q: To support the claim that the steering method controls self-reflection itself, the authors should report direct changes in judged self-reflection rates under SR-Enhanced and SR-Suppressed, and compare against length-matched or compute-matched baselines.**
>
> A: We added a direct judged self-reflection-rate analysis under different steering strengths.
> As shown in Fig. 6(a) in Appendix G, SR Enhanced increases the fraction of outputs labeled as genuine self-reflection, while SR Suppressed decreases it, establishing that the intervention modulates the intended behavior rather than only changing accuracy or length.
>
> **Q: The cosine-similarity analysis in Figure 2 left difficult to interpret.**
>
> A: We have discussed this analysis in Section 6.1, and are happy to clarify its intended interpretation.
> The cosine-similarity analysis serves two purposes.
> First, the high similarity between self-reflection vectors from MATH500 and GPQA Diamond shows that reflection-related directions are consistent across reasoning domains, supporting cross-domain transferability.
> Second, the comparison with the ''wait'' embedding is only a qualitative sanity check for surface-token dependence, not a direct baseline for vector-to-vector similarity.
> We use ''wait'' because it is the most frequent self-reflection onset token for DeepSeek models in our judge-labeled outputs.
> These two quantities answer different questions: task-to-task vector similarity tests whether the reflection direction is stable across datasets, while similarity to ''wait'' tests whether the direction collapses to a single lexical marker.
> The lower similarity to the ''wait'' embedding suggests that the reflection vector captures a broader hidden-state pattern beyond simply producing ''wait''. Moreover, the high task-to-task vector similarity suggests that the direction is not merely task-specific, but instead captures a reflection-related pattern shared across reasoning datasets.

---

> ### Author Response · Authors · 2026-05-20
>
> **Q: The budget forcing baseline could be strengthened.**
>
> A: Our budget-forcing (BF) baseline follows the original procedure from the paper (s1: Simple test-time scaling): the ''wait'' token is appended only when the model produces a short response, with the goal of preventing premature termination and forcing additional test-time computation.
> In our implementation, we define short responses as generations shorter than 8196 tokens.
> We will make this explicit in the revised manuscript.
>
> We use BF as a representative surface-token test-time scaling baseline, rather than as an exhaustively tuned prompting method.
> The comparison shows that BF often increases length substantially, but provides weaker or inconsistent gains compared with SR steering.
> Together with our verbosity-control direction experiments, this suggests that the benefit of SR steering is not explained by simply forcing longer generations.
>
> **Q: Equation 6 should be clarified.**
>
> A: In Eq. 6, the bracketed term $\langle h^{(\ell)}, v^{(\ell)} \rangle$ denotes the inner product between the current hidden state and the self-reflection vector.
> It is a scalar projection coefficient that measures how aligned the current hidden state is with the reflection direction.
> Thus, positions already aligned with the reflection direction receive a stronger update, while unrelated positions receive a weaker update.
> We have clarified this interpretation in the text around Eq. 6.
>
> **Q: The paper's claim that pretrained models already possess self-reflection capabilities is not yet convincingly supported. The strongest direct evidence for spontaneous self-reflection in the pretrained model appears to be a 0.6\% rate on MATH500. If this corresponds to only a few examples, then the result is too small to carry the broad claim that self-reflection "emerges naturally" in pretrained models. This point is especially important because the self-reflection labels are produced by an LLM-as-a-judge. The paper should show the actual labeled examples, validate the judge against human assessments, and report whether the result is reproducible across seeds, prompts, and judge models. The reflection-inducing probing result is more interesting, but I do not think it fully resolves this issue. Since the pretrained model is conditioned on a prefix generated by a fine-tuned reasoning model, the result may reflect sensitivity to local discourse patterns or near-reflection-triggering tokens in the prefix, rather than an independently available self-reflection capability. The authors should analyze how often the inserted prefix already contains reflection-adjacent cues, and how predictive those cues are of the pretrained model's subsequent reflection. Without this ablation, the 38.9\% result is difficult to interpret.**
>
> A: We added two additional analyses to address this concern.
> First, to validate label reliability, we report the judge model, prompt, decision rule, and a 50-example human validation set in Appendix C.
> The judge agrees with human labels on 50/50, 50/50, and 49/50 examples across three repeated runs, and we include representative labeled examples.
>
> Second, we added prefix-control experiments in Table 2 to test whether pretrained models can produce self-reflection specifically under reflective contexts.
> The reflective prefix yields 38.9\% judged self-reflection, and a rewritten reflective prefix gives a similar rate of 34.2\%.
> In contrast, length-matched, style-matched, and shuffled prefixes produce almost no self-reflection.
> These results show that the pretrained model can generate self-reflective continuations when given a coherent reflective prefix, but does not do so under non-reflective or corrupted prefixes.
> Thus, the effect is not explained by prefix length or generic discourse style alone.
> We also provide a representative Qwen2.5-7B self-reflection example in Appendix F.
>
> **Table 2. Control experiments for reflection-inducing prefixes on MATH500.** “Reflective” uses the original reflective CoT prefix. “Rewritten” is produced by prompting another LLM to rewrite the reflective prefix while keeping the final ten tokens unchanged. “Len.-matched” uses a non-reflective CoT prefix with comparable token length. “Style-matched” removes explicit self-reflection-inducing content, including the final ten tokens. “Shuffled” randomly shuffles the reflective prefix.
>
> | Metric | Original | Reflective | Rewritten | Len.-matched | Style-matched | Shuffled |
> |---|---:|---:|---:|---:|---:|---:|
> | Reflection Rate | 0.4 | 38.9 | 34.2 | 0.1 | 0.0 | 0.0 |
> | Acc. | 27.5 | 32.1 | 31.4 | 28.6 | 28.0 | 10.8 |

---

> ### Author Response · Authors · 2026-05-20
>
> **Q: I am not convinced that ROSCOE-SS is sufficient to support the claim that reasoning quality is preserved. A semantic coherence score may stay similar even if the model shifts toward a different reasoning strategy, relies more on shortcuts, or produces reasoning that is locally coherent but less reliable. In particular, suppressing self-reflection may cause the model to follow intuitive but sometimes misleading reasoning paths more often. Such outputs could still receive similar ROSCOE-SS scores while being meaningfully worse as reasoning traces. I would frame this as a limitation, or supplement the analysis with a process reward model, human evaluation, or correctness-conditioned analysis of the reasoning steps.**
>
> A: Thanks for the comment. We note that our main evidence for improved reasoning is the downstream Pass@1 improvement, while ROSCOE-SS is used only as a complementary diagnostic for the generated reasoning traces.
> Following reviewer's suggestion, we added a correctness-conditioned analysis in Table 3, computing ROSCOE-SS separately for examples with correct and incorrect final answers.
> The scores remain close to Vanilla in both groups, suggesting no clear degradation in the ROSCOE-SS-measured quality of reasoning traces after steering. Meanwhile, we acknowledge that ROSCOE-SS may not fully capture all aspects of reasoning quality; more direct evaluation of reasoning processes (e.g., human evaluation) could provide additional insights, but such analyses require substantial human efforts and are beyond the scope of this work. Accordingly, we have revised the manuscript to restate the result more cautiously: SR steering improves downstream task performance, while ROSCOE-SS provides supporting evidence that the generated reasoning traces are not obviously degraded, though it cannot rule out all possible changes in reasoning strategy. We have incorporated all the above discussion in the revised manuscript.
>
> **Table 3. Correctness-conditioned ROSCOE-SS analysis under self-reflection steering.** We separately report ROSCOE-SS for examples with incorrect and correct final answers.
> | Setting | ROSCOE-SS incorrect | ROSCOE-SS correct |
> |---|---:|---:|
> | Vanilla | 0.557 | 0.579 |
> | SR Enhanced | 0.551 | 0.573 |
> | SR Suppressed | 0.562 | 0.584 |

---

> > ### Comment · Reviewer_oxuz · 2026-05-20
> >
> > Thanks for all the precisions and additional results, they are quite appreciated --- good work.

---

> > > ### Author Response · Authors · 2026-05-22
> > >
> > > Thank you for taking the time to review our response.

---

> > > > ### Comment · Reviewer_oxuz · 2026-05-22
> > > >
> > > > Maybe a tiny bonus question: do you plan on making your source code (for experiments and evaluations) public?

---

> > > > > ### Author Response · Authors · 2026-05-22
> > > > >
> > > > > Thank you for the question. Yes, the source code for our experiments and evaluations is already included in the anonymized supplementary material. We will make the non-anonymized public repository available after the review period.

---

### Review · Reviewer_WTft · 2026-05-08

**Summary Of Contributions:**

The paper studies whether self-reflection in language models is something uniquely introduced by RLVR, or whether it already exists as a latent capability in pretrained models.
It introduces reflection-inducing probing, showing that pretrained models can produce self-reflective behavior much more frequently when given pre-reflection reasoning traces from fine-tuned models.
The paper also identifies a self-reflection vector in activation space and uses it to steer reflective behavior in both directions.
Enhancing this vector often improves reasoning accuracy, while suppressing it can reduce response length with limited performance loss.

Overall, the paper is well-motivated, clearly written, and solves an important question related to reasoning, RLVR, and activation steering.
The main weankness is that some of the claims, especially those about latent self-reflection mechanisms and task-agnostic control, would be more convincing with stronger controls, clearer judge validation, and more complete reproducibility details.

**Audience:**

Yes

**Audience Explanation:**

This paper has interesting findings, and should interest readers working on reasoning models, RLVR, activation steering, and model interpretability.

**Broader Impact Concerns:**

The work studies activation steering for controlling reasoning behavior.
This has useful applications for reducing inference cost and improving reasoning, but it also raises safety concerns because steering vectors may have side effects beyond the target behavior.
The paper can discuss whether enhancing or suppressing self-reflection changes refusal behavior, hallucination rate, calibration, or safety-related outputs.

**Claims And Evidence:**

Yes

**Claims Explanation:**

Overall, the main claims are supported by clear empirical evidence across multiple model families and benchmarks.
The results show that pretrained models rarely exhibit self-reflection in normal decoding but can do so much more often under reflection-inducing probing.
The hidden-state analysis and steering experiments also provide consistent evidence that reflective and non-reflective contexts are separable in activation space.

But some of the stronger interpretations can be stated more carefully.
Reflection-inducing probing may partly capture continuation of a reflective reasoning style rather than fully isolating an internal self-reflection mechanism.
And the LLM judge is vital to the labeling pipeline, but its model version, reliability, and human validation are not sufficiently reported.
The statistical analysis is also somewhat limited, especially for small benchmarks.

**Requested Changes:**

1. Provide a fuller description and validation of the LLM-as-a-judge pipeline. The paper can report the judge model, version, decoding settings, system prompt if any, and whether the labels are stable across repeated judge calls. A small human validation set with agreement statistics is also very important.
2. Add stronger control experiments for reflection-inducing probing. In particular, compare against length-matched non-reflective CoT prefixes, rewritten prefixes, shuffled or partially corrupted prefixes, and prompts that match surface style without leading to genuine revision. This can help separate latent self-reflection from simple continuation of a reflective writing pattern.
3. Add clarification on how the self-reflection vectors are estimated, how the data are split, and how steering coefficients ($\alpha$) and injection layers are selected.
4. Add statistical reporting, such as confidence intervals, paired bootstrap tests, or item-level comparisons, particularly for small datasets such as AIME 2024. The current reporting of means and standard deviations over donor models is useful but does not fully capture item-level uncertainty.
5. Report wall-clock latency, throughput, and memory overhead in addition to output length.
6. Add more discussion of negative or mixed results, such as the drop on GPQA Diamond for Qwen2.5-7B under SR Enhanced (Table 1).
7. Add a paired comparison across the same backbone before and after SFT, distillation, or RLVR when possible.

---

> ### Author Response · Authors · 2026-05-20
>
> **Q: Provide a fuller description and validation of the LLM-as-a-judge pipeline.**
>
> A: We thank the reviewer for the suggestion.
> We have added a detailed description and validation of the LLM-as-a-judge pipeline in Appendix C, including the judge model and version, decoding settings, the full judging prompt, the decision rule, and repeated-call stability.
>
> We also added a small human-labeled validation set with 50 examples: 25 manually identified genuine self-reflective responses from RLHFlow/Deepseek-MATH500-Test and 25 non-reflective passages from HuggingFaceFW/fineweb-edu.
> Across three repeated judge runs, the judge agrees with human labels on 50/50, 50/50, and 49/50 examples, respectively.
> This suggests that the judge labels are stable and sufficiently aligned with human judgments for our analysis.
>
> **Q: Add stronger control experiments for reflection-inducing probing.**
>
> A: We have added prefix-control experiments in Table 1.
> We compare the original reflective prefix with rewritten reflective prefixes, length-matched non-reflective CoT prefixes, surface-style matched prefixes, and shuffled prefixes.
>
> The results show that only prefixes preserving reflection-relevant content substantially increase judge-labeled self-reflection and accuracy.
> In contrast, length-matched and surface-style matched prefixes produce almost no self-reflection, while shuffled prefixes substantially hurt performance.
> These controls suggest that the effect depends on coherent reflection-inducing content.
>
> **Table 1. Control experiments for reflection-inducing prefixes on MATH500.** “Reflective” uses the original reflective CoT prefix. “Rewritten” is produced by prompting another LLM to rewrite the reflective prefix while keeping the final ten tokens unchanged. “Len.-matched” uses a non-reflective CoT prefix with comparable token length. “Style-matched” removes explicit self-reflection-inducing content, including the final ten tokens. “Shuffled” randomly shuffles the reflective prefix.
>
> | Metric | Original | Reflective | Rewritten | Len.-matched | Style-matched | Shuffled |
> |---|---:|---:|---:|---:|---:|---:|
> | Reflection Rate | 0.4 | 38.9 | 34.2 | 0.1 | 0.0 | 0.0 |
> | Acc. | 27.5 | 32.1 | 31.4 | 28.6 | 28.0 | 10.8 |
>
> **Q: Add clarification on how the self-reflection vectors are estimated**
>
> A: These details are already described in the manuscript, but we are happy to clarify them here in one place.
> The self-reflection vector is estimated as a difference-in-means direction between reflection-inducing and non-reflection-inducing hidden states, using 100 examples for each set.
> For data splitting, we use 10\% of the data as a validation set to select the steering coefficient.
> For intervention selection, we perform a grid search over $\alpha \in [-1,1]$ and compare candidate injection layers on the validation set.
> The final steering coefficient and injection layer are then selected according to validation performance.
> Additional ablations on steering strength and injection layer are provided in Section 6.5.
>
> **Q: Add statistical reporting**
>
> A: We added Table 2, which reports Pass@1 together with paired bootstrap 95\% confidence intervals for differences relative to Vanilla.
> The bootstrap is performed over evaluation items, preserving the pairing between Vanilla and each intervention on the same problem.
> The resulting intervals show that BF does not provide a reliable improvement over Vanilla, while SR Enhanced gives a modest positive gain on DeepSeek-R1-7B and a substantially larger positive gain on Qwen2.5-7B.
>
> **Table 2. AIME 2024 uncertainty analysis over four repeated evaluation runs.** We report the mean Pass@1 and average generation length with standard deviations across four runs. For each method, Δ denotes the difference relative to Vanilla. The 95% confidence intervals for Δ are computed by paired bootstrap resampling over evaluation items, where each item is represented by its average correctness over the four runs.
> | Model | Method | AIME 2024 Pass@1 | Avg. Len. | Δ vs. Vanilla [95% CI] |
> |---|---|---:|---:|---:|
> | DeepSeek-R1-7B | Vanilla | 55.8±3.1 | 4737±494 | -- |
> |  | BF | 54.5±2.6 | 9844±1656 | -1.3 [-3.2, +2.4] |
> |  | SR Enhanced | 58.2±2.8 | 6352±1249 | +2.4 [+0.7, +6.9] |
> |  | SR Suppressed | 52.9±3.3 | 3692±685 | -2.9 [-3.6, +0.5] |
> | Qwen2.5-7B | Vanilla | 3.9±0.4 | 2285±297 | -- |
> |  | BF | 5.3±0.9 | 4526±401 | +1.4 [-1.0, +3.7] |
> |  | SR Enhanced | 16.1±2.4 | 2941±486 | +12.2 [+10.8, +20.4] |

---

> > ### Author Response · Authors · 2026-05-20
> >
> > **Q: Report wall-clock latency, throughput, and memory overhead in addition to output length.**
> >
> > A: We now report end-to-end wall-clock latency, decoding throughput, and peak GPU memory usage under the same decoding setup as the main experiments.
> > All measurements are obtained on the same hardware, with batch size 1 and greedy decoding.
> > As shown in Table 3, self-reflection steering increases wall-clock latency mainly because it changes the number of generated tokens.
> > The per-token decoding throughput remains similar to the vanilla model, and the peak-memory overhead is small because the steering intervention only adds a fixed direction at the selected layer rather than changing model parameters or storing additional KV-cache states.
> > Thus, the computational cost of SR steering is primarily mediated by longer generations, while the activation-space intervention itself introduces only modest runtime and memory overhead.
> >
> > **Table 3. Efficiency analysis on two representative 7B models.** Latency denotes average end-to-end wall-clock generation time per example. Throughput is measured as generated tokens per second. Peak memory is the maximum allocated GPU memory during generation, and Δ Mem. is measured relative to the vanilla setting for the same model. All measurements use the same hardware, batch size, maximum generation length, and greedy decoding.
> > | Model | Setting | Avg. Len. | Latency (s) | Throughput (tok/s) | Peak Mem. (GB) | Δ Mem. |
> > |---|---|---:|---:|---:|---:|---:|
> > | DeepSeek-R1-7B | Vanilla | 3585 | 69.1 | 51.9 | 20.4 | -- |
> > |  | BF | 7111 | 141.4 | 50.3 | 21.1 | +0.7 |
> > |  | SR Enhanced | 8959 | 178.5 | 50.2 | 21.5 | +1.1 |
> > | Qwen2.5-7B | Vanilla | 1294 | 24.6 | 52.6 | 19.8 | -- |
> > |  | BF | 3986 | 78.6 | 50.7 | 20.5 | +0.7 |
> > |  | SR Enhanced | 2671 | 52.9 | 50.5 | 20.2 | +0.4 |
> >
> > **Q: Add more discussion of negative or mixed results**
> >
> > A: We have added discussion of negative and mixed results in the revised discussion.
> > In particular, SR Enhanced is not uniformly beneficial across all model-task pairs; for example, Qwen2.5-7B shows a drop on GPQA Diamond.
> > Our interpretation is that self-reflection steering is most helpful when additional deliberation can correct intermediate reasoning errors, but it may hurt when the task is more knowledge-intensive or when stronger reflection causes the model to over-elaborate without adding useful evidence.
> >
> > **Q: Add a paired comparison across the same backbone**
> >
> > A: We agree that an exact paired comparison before and after SFT, distillation, or RLVR would be the cleanest way to isolate the effect of post-training.
> > However, this is not feasible in our current setting because we rely on publicly released checkpoints, for which the exact corresponding pretraining checkpoints and intermediate post-training stages are generally unavailable.
> > Instead, our experiments compare pretrained and post-trained models across several model families and show that self-reflection-related representations are linearly accessible in both, while post-trained models express this behavior much more frequently.
> >
> > **Q: The work studies activation steering for controlling reasoning behavior. This has useful applications for reducing inference cost and improving reasoning, but it also raises safety concerns because steering vectors may have side effects beyond the target behavior. The paper can discuss whether enhancing or suppressing self-reflection changes refusal behavior, hallucination rate, calibration, or safety-related outputs.**
> >
> > A: We have expanded the broader impact discussion in Appendix D to discuss possible side effects of activation steering beyond reasoning efficiency.

---

> > > ### Comment · Reviewer_WTft · 2026-06-07
> > >
> > > Thanks to the authors for the detailed response and additional experiments. The added judge validation, prefix controls, statistical analysis, and efficiency analysis address most of my main concerns. I still think there are some limitations in the judge validation and in the breadth of the task and model scale, but these do not affect my overall leaning toward acceptance.

---

> > > > ### Author Response · Authors · 2026-06-08
> > > >
> > > > Thank you for the thoughtful assessment and for recognizing the added experiments. We are glad that the response and additional analyses addressed most of your main concerns.

---

### Review · Reviewer_YWp7 · 2026-05-09

**Summary Of Contributions:**

The paper's central claim is that self-reflection in LLMs exists as a latent capability in pretrained models, is geometrically encoded in hidden states, and can be controlled bidirectionally via a single activation-space direction.

My assessment: partially achieved. The practical steering method works, generalizes across models, and gives you a real accuracy-efficiency knob. But the paper does not convincingly establish that (a) pretrained models have "latent self-reflection capability" as opposed to pattern-completion ability, (b) the extracted direction specifically encodes self-reflection rather than something like general deliberation intensity, or (c) the causal pathway goes through actual reflection rather than some correlated mechanism.

Strengths I want to highlight:

- The methodology is clean and the recipe is practical — difference-in-means followed by projected steering at a chosen layer. Someone could implement this in an afternoon.
- Evaluation breadth is good: four fine-tuned models (DeepSeek-R1 at 1.5B and 7B, OLMo2-13B, OLMo3-7B-Think), three pretrained models (Qwen2.5-1.5B, Qwen2.5-7B, Llama 3.1-8B), spanning four different model families.
- The cross-domain transfer result is genuinely interesting and well-demonstrated.
- Budget forcing comparison shows their method achieves better accuracy per token than naive "wait" insertion.
- The ROSCOE-SS evaluation was a smart inclusion — it preempts the "are you just corrupting the reasoning chain?" objection.
- Table 4's comparison of extraction methods (difference-in-means, additive steering, PCA, contrastive PCA) is informative for the steering-vector community specifically.

Weaknesses that need addressing:

- No verification that steering changes actual reflection frequency (the judge-labeled rate is never measured as a function of alpha). This is the single biggest gap.
- No control directions tested. Without this, we can't say the effect is reflection-specific.
- The "latent capability" claim conflates in-context pattern completion with dormant metacognition.
- Linear separability is presented as surprising when it's expected under any autoregressive model that predicts its own continuations.
- All benchmarks are math/science. The "diverse tasks" claim is unsupported.
- Judge model unspecified. Variance reporting inconsistent across Apt/Aft.
- The abstract's "13.7%" highlight is from pretrained models with low baselines; deployed fine-tuned models see 1-5pp gains. The framing misleads.
- Potential circularity: same judge defines what reflection is and (implicitly) evaluates whether the method works.

Overall, I see this as a solid applied contribution — inference-time steering for the accuracy-efficiency tradeoff — that has been dressed up as a deeper mechanistic insight than the evidence supports. The engineering is good. The science needs more work.

**Additional Comments:**

Some things I appreciated that I haven't mentioned yet:

The projected steering in Eq. 6 (where the perturbation scales by how aligned the current hidden state is with the reflection direction) is a nice design choice. It's more principled than dumping a constant vector in at every position, and the comparison in Table 4 confirms it matters. I'd like to see this discussed more — why does projected outperform additive? What does that tell us about how reflection-relevant information is distributed across positions?

The budget forcing comparison was smart to include. It makes the concrete case that you can't get the same effect by just stuffing "wait" tokens into the generation — the vector steering gives better accuracy per token spent. That's a clean practical demonstration.

Table 5's qualitative examples helped me understand what the judge is labeling. The OLMo "But this step assumes independence, which is not valid..." example is clearly genuine reflection. The examples make me trust the labeling more than the methodology section alone did.

Questions I'm left with:

- If you apply positive alpha to a trivial problem (2+3), does the model produce unnecessary deliberation? Is there a difficulty threshold below which enhanced reflection hurts?
- The optimal alpha is 0.03 for DeepSeek-R1-1.5B. Is that transferable across models, or does every model need its own grid search? This matters a lot for practical deployment.
- Table 6 shows different models have different characteristic reflection tokens (OLMo prefers "But"/"hold," Qwen uses "hmm"). Do the corresponding vectors point in qualitatively different directions, or do they all converge to a shared subspace?
- How does this compare to just prompting "Double-check your reasoning before answering" vs. "Answer directly"? If prompting gets you 80% of the effect without touching hidden states, the added complexity of this method needs justification.

In terms of positioning relative to concurrent work: Wang et al. (2025) "Wait, we don't need to 'wait'!" is working on essentially the same suppression problem from a different angle. A direct comparison would strengthen this paper. The broader test-time compute literature (best-of-N, majority voting, process reward models) provides context for where this method sits — it's cheaper than sampling multiple times but more invasive than pure prompting. That positioning would help readers.

Finally, the paper would benefit from connecting to the linear representation hypothesis literature more explicitly. Arditi et al. (2024) showed refusal is mediated by a single direction; this paper suggests deliberation/reflection might be too. What's the emerging picture? Are all interesting behavioral modes linearly represented, or is this specific to certain types of behavior? That's a question worth at least one paragraph of discussion.

Scaling-wise, all models in this paper are ≤13B parameters. I'm curious whether reflection vectors exist and work the same way in 70B+ models. That's probably future work, but worth flagging.

**Audience:**

Yes

**Audience Explanation:**

TMLR readers working on efficient inference, test-time compute allocation, or reasoning model deployment will find direct value here. The interpretability community will appreciate the methodology but want deeper analysis. Researchers studying emergent capabilities will be interested in the question even if the answer is incomplete.

The scope is appropriate for TMLR, which accepts incremental but correct work. The practical method meets that bar easily. The scientific framing reaches further than the evidence allows, but that's fixable — the authors don't need to restructure the paper, just calibrate their language.

The breadth limitation is real: someone building a chatbot or a code assistant can't tell from this paper whether reflection steering would help them. Testing on even one non-math benchmark would expand the audience considerably.

**Broader Impact Concerns:**

The positive side is straightforward: a tunable knob for reasoning effort at inference time helps with deployment costs, and the method is transparent (a direction and a scalar) rather than a black-box prompt hack. Understanding what pretraining gives you for free could inform how we do fine-tuning.

The concerning side deserves more discussion than the paper provides. Suppressing self-reflection means producing a model that doesn't pause to reconsider — in medical, legal, or financial contexts, that's potentially dangerous. More subtly, if models sometimes use reflection-like patterns as informal safety checks ("Wait, should I really be helping with this?"), suppressing that direction might weaken guardrails in ways that aren't immediately obvious.

On the flip side, enhancing reflection could help a model construct more elaborate incorrect rationalizations — longer doesn't always mean better, and a model that "reflects" more confidently on wrong reasoning could be harder to catch.

None of this is disqualifying, but the paper should acknowledge these dual-use considerations rather than presenting the method as purely beneficial.

**Claims And Evidence:**

Yes

**Claims Explanation:**

**What holds up well under scrutiny:** The steering effect itself is real and reproducible across models (Table 1). The cross-task transfer with cosine >0.9 between domains is well-documented (Figure 2, Table 3). Difference-in-means beats alternatives for vector extraction (Table 4). Middle-layer injection outperforms early and late layers (Figure 2, right panel). ROSCOE-SS stability confirms the intervention doesn't destroy reasoning coherence (Table 2). The vector is not just the "wait" embedding in disguise (cosine <0.25 with "wait").

**What's weakly supported:** The "latent capability in pretrained models" interpretation (the alternative explanation of pattern completion is never ruled out). That the vector encodes self-reflection specifically rather than a correlated property. That improved accuracy *comes from* more reflection rather than more computation.

**What's not supported at all:** That the intervention mechanism is reflection-specific (no control baselines exist). That >99% linear separability says something specific about self-reflection vs. being a trivial prediction-objective consequence. That this constitutes "precise behavioral control" when per-model hyper parameter search is required.

To close these gaps, the authors need:
1. The decisive experiment — run the judge on outputs at different alpha values and show a dose-response curve between steering strength and reflection frequency. This is the one thing that would convert me.
2. Control directions — random vectors matched in norm, a "verbosity" vector (long vs short response hidden states), maybe a "correctness" vector. If the reflection vector doesn't qualitatively outperform these, the narrative collapses.
3. Per-problem breakdown — does enhanced reflection specifically rescue problems where the base model made a correctable error? Or does it just help everywhere uniformly (suggesting it's generic compute increase)?
4. At least one non-math evaluation — commonsense reasoning, code, multi-hop QA — anything to test the "diverse tasks" claim.
5. Some acknowledgment that pattern completion and latent metacognition are distinct hypotheses, even if the paper can't definitively choose between them yet.

**Requested Changes:**

**[Critical] Verify that steering actually changes reflection frequency**

The whole paper claims this vector "controls self-reflection." Nowhere do the authors check whether alpha actually modulates how often the judge labels outputs as genuinely reflective. They show accuracy changes and length changes, but these could arise from many mechanisms that aren't reflection-specific.

This matters because the entire scientific contribution rests on an assumed causal link: steering with this vector → model reflects more → accuracy improves. If the middle step doesn't hold, the paper is really about "a useful direction that makes models deliberate more" — still publishable, but a fundamentally different claim.

What I need: run the judge (same one used for vector construction) on model outputs generated under different alpha values. Plot reflection rate vs. alpha. If there's a clear dose-response — more alpha means proportionally more genuine reflection — the paper's story checks out. If reflection rate barely changes while length and accuracy do, the vector is controlling something else, and the framing needs to change.

This experiment could be done in a few days with existing infrastructure. It's not a high bar.

**[Critical] Test against control directions**

No control directions are tested anywhere in the paper. This is a glaring omission for a paper that claims specificity of its intervention.

I need to see: (1) random directions matched in norm (averaged over at least 5 draws), (2) a "verbosity" direction constructed from hidden states of long vs. short responses, and optionally (3) a "correctness" direction from hidden states preceding correct vs. incorrect answers.

If the reflection vector produces roughly the same accuracy-length curve as a verbosity vector, then what you've found is a verbosity dial, not a reflection controller. If it produces clearly distinct behavior — say, improving accuracy without proportionally increasing length, or showing the dose-response with judge labels that controls don't — then the specificity claim is justified.

Without this comparison, the paper is missing the single most obvious alternative hypothesis.

**[Major] The "latent capability" claim needs either evidence or reframing**

The paper's marquee finding — pretrained models showing 38.9% self-reflection under probing — has a simple alternative explanation: the model is doing in-context pattern completion. You fed it a chain-of-thought that's obviously building toward a self-correction, and it continued the pattern. Any halfway decent language model would do this. That's not evidence of a dormant metacognitive capability; it's evidence that self-reflection patterns exist in the training data.

The authors need to either: (A) demonstrate that these induced reflections are functional — they actually identify real errors and lead to correct revisions at above-chance rates — or (B) soften the claim to something like "pretrained models can produce reflection-patterned text given appropriate context, suggesting these patterns are represented in the pretraining distribution, though whether this constitutes genuine metacognitive capability remains open."

**[Major] Contextualize the linear separability finding**

The >99% SVM result in Section 6.4 is presented as evidence that pretrained models "encode self-reflection internally." But any autoregressive model must represent different upcoming continuations differently — that's literally the training objective. This finding, without a comparison to other contrasts, doesn't tell us anything specific about self-reflection.

Either compare against at least one control contrast (e.g., separating "next clause is a question" from "next clause is a statement" at the same token positions), or reposition this as "the direction is linearly accessible, which is a precondition for our steering to work" rather than evidence of a special self-reflection encoding.

**[Major] Include at least one non-math/science benchmark**

MATH-500, AIME 2024, and GPQA Diamond are the only benchmarks. The paper claims the mechanism "transfers robustly across diverse tasks." Two math competitions and one science QA set is not diverse. Self-reflection in code debugging, commonsense reasoning, or multi-hop factual QA may look very different.

Testing the same vector on ARC, StrategyQA, or HumanEval would either confirm universality (strengthening the paper) or reveal math-specificity (still useful, but the claims need scoping). Either way it's informative.

**[Minor] Variance reporting for fine-tuned models**

Table 1 gives error bars for pretrained models but only point estimates for fine-tuned ones. If the fine-tuned models use greedy decoding, just say so. If they don't, the +3.3pp gain for DeepSeek-R1-1.5B (84.1→87.4) could easily be within noise.

**[Minor] Name the judge model and validate it**

The entire pipeline starts with LLM judge labels. Which model is judging? This isn't stated anywhere. Different judges have different thresholds. If the judge over-identifies reflection in contexts that happen to correlate with, say, longer reasoning chains, then the extracted vector might encode "long reasoning" rather than "genuine reflection."

I'd like to see: (1) the judge model named, (2) agreement with human labels on 50-100 examples, and (3) brief discussion of whether the circularity (same judge for extraction and implicit evaluation) is a concern.

**[Minor] How many samples for vector construction?**

Nowhere in the paper can I find how many RIT/NIT hidden states were collected. A practitioner wanting to replicate this needs to know: does this work with 50 samples? 500? A quick ablation (steering performance vs. sample count) would be very helpful.

**[Minor] Tone down the overstating**

"Remarkably" appears where results are interesting but not remarkable. "Reveals the hidden capacity" should be "suggests" or "is consistent with." "Strong evidence that self-reflection capabilities develop during pretraining" is too strong for what's shown. The work is good enough to speak for itself without the hard sell, and overclaiming makes reviewers (me included) scrutinize harder.

---

> ### Author Response · Authors · 2026-05-19
>
> **Q: whether steering changes reflection frequency**
>
> A: We thank the reviewer for pointing out the missing intermediate validation.
> Following this suggestion, we added an $\alpha$ analysis on MATH500 to test whether steering changes the frequency of judge-labeled self-reflection, and include the full results and discussion in Appendix G.
> Using the same steering-strength grid as in the ablation study, we generate outputs from DeepSeek-R1-1.5B and Qwen2.5-1.5B under different $\alpha$ values, and then apply the same LLM judge with a fixed prompt and decision rule.
> As shown in Fig.6(a), reflection rate changes systematically with steering strength: positive $\alpha$ increases the proportion of genuinely self-reflective outputs, while negative $\alpha$ suppresses it.
> This directly validates the intermediate step assumed in our paper: $\alpha$ modulates judge-labeled self-reflection frequency, not only accuracy or generation length.
>
> **Q: Test against control directions**
>
> A: Following the reviewer's suggestion, we added control-direction experiments on MATH500.
> We compare the self-reflection (SR) direction against norm-matched random directions averaged over five draws, a verbosity direction from long-vs.-short responses, and a correctness direction from correct-vs.-incorrect responses, using the same intervention setup on DeepSeek-R1-7B and Qwen2.5-7B.
> As shown in Table 1, random and verbosity directions substantially increase generation length but do not improve PASS@1, while the correctness direction gives only small or inconsistent gains.
> By contrast, the SR direction gives the strongest improvement, especially on Qwen2.5-7B ($44.8 \rightarrow 56.6$ PASS@1).
> Thus, SR steering is not merely a verbosity dial; it improves reasoning performance in a way that cannot be explained by increased response length alone.
> **Table 1. Control-direction comparison on MATH500.** Random directions are norm-matched to SR and averaged over five independent draws. The verbosity direction is constructed from long-vs.-short responses, where long responses are defined as  outputs longer than 4K tokens for DeepSeek-R1-7B and longer than 2K tokens for Qwen2.5-7B.
>
> | Model | Metric | None | Random | Verbosity | Correctness | SR |
> |---|---|---:|---:|---:|---:|---:|
> | DeepSeek-R1-7B | PASS@1 | 92.7±1.9 | 80.3±7.4 | 87.8±3.6 | 91.1±2.2 | 93.5±1.2 |
> |  | Len. | 3585±293 | 7312±1418 | 6124±1206 | 4718±537 | 8959±916 |
> | Qwen2.5-7B | PASS@1 | 44.8±1.9 | 36.9±6.8 | 41.7±3.9 | 46.2±2.5 | 56.6±2.7 |
> |  | Len. | 1294±15 | 2811±392 | 5476±489 | 2093±301 | 2671±462 |
>
>
> **Q: The "latent capability" claim**
>
> A: Thank you for the question and comment. We first note that, following another comment from reviewer YWp7, we have added prefix-control experiments in Table 2.
> The results show that the reflective CoT prefix substantially increases both judge-labeled reflection rate and accuracy, while a rewritten reflective prefix produces a similar effect.
> In contrast, length-matched non-reflective and surface-style matched prefixes yield almost no self-reflection and only minor accuracy changes.
> The shuffled prefix further reduces accuracy, suggesting that arbitrary length or surface continuation is not sufficient; the prefix must preserve coherent reflection-relevant content.
> These controls indicate that the induced behavior depends meaningfully on prefix content.
>
> That said, it is not our intent in this work to distinguish between metacognitive capability and pattern matching, as such a distinction would likely require broader theoretical and interdisciplinary perspectives (e.g., cognitive science or philosophy of mind), which are beyond both our expertise and the scope of this paper. Following the reviewer’s suggestion, we have revised our wording from ``latent self-reflection capability'' to the more conservative claim that "pretrained models can produce self reflection under appropriate contexts, suggesting that reflection-related patterns are represented during pretraining, whether this should be interpreted as genuine metacognitive capability remains further investigation and is beyond the scope of this paper". Specifically, we have added this discussion in footnote 1 of the revision.
>
> **Table 2. Control experiments for reflection-inducing prefixes on MATH500.** “Reflective” uses the original reflective CoT prefix. “Rewritten” is produced by prompting another LLM to rewrite the reflective prefix while keeping the final ten tokens unchanged. “Len.-matched” uses a non-reflective CoT prefix with comparable token length. “Style-matched” removes explicit self-reflection-inducing content, including the final ten tokens. “Shuffled” randomly shuffles the reflective prefix.
>
> | Metric | Original | Reflective | Rewritten | Len.-matched | Style-matched | Shuffled |
> |---|---:|---:|---:|---:|---:|---:|
> | Reflection Rate | 0.4 | 38.9 | 34.2 | 0.1 | 0.0 | 0.0 |
> | Acc. | 27.5 | 32.1 | 31.4 | 28.6 | 28.0 | 10.8 |

---

> ### Author Response · Authors · 2026-05-20
>
> **Q: Contextualize the linear separability finding**
>
> A: Following the suggestion, we added control-probe experiments on DeepSeek-R1-7B using the same probing setup as in main content.
> Specifically, we compare the self-reflection contrast with two generic control contrasts: whether the next continuation is a question vs. a statement, and whether the response is long vs. short.
> As shown in Fig.6(b), these control contrasts remain close to chance across layers, in contrast to the $>99\%$ separability observed for the self-reflection labels.
>
> **Q: Include at least one non-math/science benchmark**
>
> Following the suggestion, we added a coding benchmark evaluation on HumanEval using DeepSeek-R1-7B and Qwen2.5-7B.
> The HumanEval results show that self-reflection steering has a positive effect on code-generation performance, suggesting that the learned direction is not limited to math-style reasoning.
>
> **Table. HumanEval results for self-reflection steering under greedy decoding.** We report pass@1 and average generation length for baseline, self-reflection enhancement, and self-reflection suppression on DeepSeek-R1-7B and Qwen2.5-7B.
>
> | Model | Setting | Pass@1 | Avg. Len. |
> |---|---|---:|---:|
> | DeepSeek-R1-7B | Baseline | 93.5 | 6943 |
> |  | SR Enhance | 96.7 | 8214 |
> |  | SR Suppress | 91.4 | 4341 |
> | Qwen2.5-7B | Baseline | 82.8 | 3493 |
> |  | SR Enhance | 89.5 | 6114 |
>
> **Q: Variance reporting for fine-tuned models**
>
> Both $A_{\mathrm{ft}}$ and $A_{\mathrm{pt}}$ results in Table 1 are obtained using greedy decoding.
> For $A_{\mathrm{pt}}$ models, the error bars come from constructing steering vectors with prefixes generated by four different $A_{\mathrm{ft}}$ models, not from decoding randomness.
> For $A_{\mathrm{ft}}$ models, we use a fixed validation-selected steering configuration and report point estimates.
> We have clarified this in the revised manuscript.
>
> **Q: Name the judge model and validate it**
>
> We have added a detailed description of the judge pipeline in Appendix C.
> Briefly, we use *Gemini-3-Flash* as the judge.
> Since manually labeled self-reflection data are scarce, we construct a lightweight human-label validation set ourselves.
> Specifically, we manually identify 25 genuine self-reflective responses from RLHFlow/Deepseek-MATH500-Test as positive examples, and use the first 25 ordinary web-text passages from HuggingFaceFW/fineweb-edu as non-reflective negative examples.
> Across three repeated judge runs, the judge correctly labels all 25 negative examples in every run and labels 25, 25, and 24 of the 25 positive examples correctly.
> These results suggest that the judge is sufficiently reliable for separating genuine reflection from surface markers.
>
> We also discuss the possible circularity of using the same judge for identifying reflection instances and measuring reflection rates. Since the judge performs a constrained evidence-identification task with a fixed definition of genuine self-reflection, and our human-label validation shows high agreement, we do not view this as a severe source of bias. However, we acknowledge that the exact reflection rates are judge-dependent. Importantly, our main downstream evaluations use independent metrics: task accuracy and generation length, rather than judge scores alone.
>
> **Q: How many samples for vector construction?**
>
> A: In the original submission, we used $100$ construction samples to build the RIT/NIT difference-in-means direction, but this was not stated clearly. We have now added this detail to the method section.
> Following the reviewer's suggestion, we also added a sample-count ablation.
> We construct the self-reflection direction using $N \in \{25,50,100,250,500\}$ self-reflection samples, with the same construction and evaluation protocol otherwise fixed.
> As shown in Table 3, the steering effect is already visible with a small number of samples and becomes more stable but less powerful as the number of construction samples increases.
> This suggests that the method does not rely on a very large labeled set.
>
> **Table 3. Sample-count ablation on Qwen2.5-7B over 50 held-out MATH500 questions.**
>
> | N | No steering | 25 | 50 | 100 | 250 | 500 |
> |---|---:|---:|---:|---:|---:|---:|
> | SR rate | 0.4 | 30.9 | 29.8 | 31.6 | 25.2 | 24.9 |
> | Acc. | 44.8 | 52.1 | 57.2 | 56.6 | 54.3 | 51.6 |
> | Avg. len. | 1294 | 1973 | 2340 | 2671 | 2226 | 2954 |
>
> **Q: Tone down the overstating**
>
> A: In the revised manuscript, we have toned down the language throughout the discussion.
>
> **Q: Broader Impact Concerns**
>
> A: Thank you for raising this important point. In the revision, we have expanded the broader impact discussion in Appendix~D to include discussions along this line.

---

> ### Author Response · Authors · 2026-05-20
>
> **Q: why does projected outperform additive**
>
> A: Our interpretation is that projected steering acts as a simple state-dependent adaptive controller.
> In Eq. 6, the inner product $\langle h^{(\ell)}, v^{(\ell)} \rangle$ is a scalar projection coefficient, so the update can be rewritten as an additive intervention with an effective coefficient $\alpha_{\mathrm{eff}}(h)=\alpha\langle h^{(\ell)},v^{(\ell)}\rangle$.
> Thus, the intervention strength varies across positions depending on how strongly the current hidden state is already aligned with the reflection direction.
> We have added a detailed discussion of this interpretation in Appendix E.
>
> **Q: If you apply positive alpha to a trivial problem (2+3), does the model produce unnecessary deliberation?**
>
> A: For very easy problems, increasing the reflection coefficient may allocate unnecessary test-time computation and can potentially hurt the accuracy--efficiency trade-off.
> Our intended use is therefore as a controllable inference-time knob, rather than a fixed decoding rule applied uniformly to all inputs.
> We have added difficulty-adaptive calibration as a limitation and future direction.
>
> **Q: The optimal alpha is 0.03 for DeepSeek-R1-1.5B. Is that transferable across models, or does every model need its own grid search?**
>
> A: The optimal $\alpha$ is not directly transferable across models.
> In our current implementation, each model requires a small held-out grid search, because activation scales and baseline reflection rates differ across architectures and training recipes.
> A natural future direction is adaptive calibration, where the steering strength is selected based on input difficulty or model confidence, so that easy problems receive little or no additional reflection while harder problems receive stronger steering.
>
> **Q: Do the corresponding vectors (OLMo prefers "But"/"hold," Qwen uses "hmm") point in qualitatively different directions, or do they all converge to a shared subspace?**
>
> A: Each model has its own representation space, tokenizer, and selected steering layer, so the raw reflection vectors are not directly comparable across model families without an additional alignment procedure.
>
> Also, we avoid constructing the direction from a single reflection token such as ''wait'', ''hmm'', or ''but''.
> A single-token difference-in-means direction could capture token-specific lexical features rather than a higher-level reflection concept.
> Our LLM-as-a-judge pipeline is intended to collect diverse self-reflective continuations, making the resulting direction less tied to any one surface marker.
>
> We therefore interpret Table 6 in main paper as evidence that different models realize reflection with different surface forms, while each model still admits a linearly accessible reflection-related direction.
> Whether these directions align to a shared cross-model subspace remains an open question requiring representation alignment or subspace-level analysis.
>
> **Q: How does this compare to just prompting "Double-check your reasoning before answering" vs. "Answer directly"?**
>
> A: Our budget-forcing baseline is intended to capture this family of surface-level interventions, which is closely related to prompting the
> model to double-check its reasoning.
> However, this strategy is weaker than activation steering. Our budget-forcing comparison shows that steering achieves a better accuracy--token trade-off, especially for pretrained models.
> Moreover, activation steering enables bidirectional control: positive coefficients enhance reflection, while negative coefficients suppress it.
> It therefore serves as a tool for probing internal reflection-related representations, justifying the added white-box complexity.
>
> **Q: Concurrent work && connecting to the linear representation hypothesis literature more explicitly && Larger Models**
>
> A: We have added the discussion in Related Works and Conclusion to clarify the positioning of our work.
> We now discuss Wang et al. (2025) as a complementary suppression approach, whereas our method intervenes on an internal activation-space direction and enables continuous bidirectional control.
> We also position our method relative to broader test-time compute: it is cheaper than multi-sample methods such as best-of-$N$ or majority voting, but more invasive than prompting due to its white-box activation access.
>
> We further connect our findings to the linear representation hypothesis, while clarifying that we do not claim self-reflection is fully one-dimensional or that all behavioral modes are linearly controllable.
> Recent work such as [1] further suggests that steering may depend on richer geometric structure beyond a single global direction.
> Understanding whether reflection is better modeled as a direction, subspace, or nonlinear structure is an important future direction.
> We also explicitly list evaluation on 70B+ models as future work.
>
> [1] The Cylindrical Representation Hypothesis for Language Model Steering

---

> > ### Comment · Reviewer_YWp7 · 2026-05-26
> > **The response presents convincing new experiments, but several key ones (control-direction comparison, HumanEval, prefix-controls, sample-count ablation) appear only in the response letter not in the revised manuscript. The dose-response curve and linear-probe controls are in the paper and look good. The control-direction table must be added to the manuscript; it's the decisive specificity evidence**
> >
> > **On the dose-response curve (my critical ask #1):**
> >
> > Figure 6(a) in Appendix G shows reflection rate as a function of alpha for DeepSeek-R1-Distill-Qwen-1.5B and Qwen2.5-1.5B, using the same alpha grid as the main ablation (-1 to 1, logarithmic spacing). The relationship is clearly monotonic: positive alpha increases judge-labeled genuine reflection, negative alpha suppresses it. For the pretrained model the curve goes from near-zero at baseline to substantial reflection under positive steering. This directly validates what I asked for the vector modulates actual self-reflection frequency, not just output length or accuracy indirectly.
> >
> > Two observations. First, this is shown on two 1.5B-scale models from the same family (both Qwen-based). The trend is clear at this scale, though showing it on the 7B models used elsewhere would complete the picture. Second, the curve confirms the vector modulates reflection *rate*, but doesn't tell us whether those additional reflections are *functional* do they catch real errors? The paper's causal story assumes alpha → more reflection → better accuracy; this experiment confirms the first link. The second link is supported indirectly by the control-direction results in the response letter.
> >
> > **On control directions (my critical ask #2):**
> >
> > The response letter presents a table comparing SR against norm-matched random directions (5 draws), a verbosity direction, and a correctness direction on DeepSeek-R1-7B and Qwen2.5-7B. The reported numbers are decisive: random directions harm accuracy substantially, verbosity increases length without helping accuracy, and only SR gives meaningful gains (particularly +11.8pp on Qwen2.5-7B).
> >
> > However, this experiment does not appear anywhere in the revised manuscript. I searched for the specific numbers, for terms like "random direction," "verbosity direction," "correctness direction," "control direction," and "norm-matched" none appear in the paper. This is a problem. The control-direction comparison is the single most important result for ruling out the alternative hypothesis that the SR vector is just a verbosity or compute dial. It belongs in the main text, not buried in a response letter that future readers won't see. I cannot fully endorse the paper's specificity claim as currently written in the manuscript, because the manuscript itself contains no evidence that competing directions fail to produce similar effects.
> >
> > **On the linear-probe controls (Figure 6b):**
> >
> > This IS in the paper and it's genuinely informative. Probes for question-vs-statement and long-vs-short contrasts remain near chance (0.50) across all 28 layers on DeepSeek-R1-7B, while the self-reflection probe exceeds 0.99. I was wrong in my initial review to dismiss the >99% separability as trivially expected these controls demonstrate it's specific to self-reflection.
> >
> > One design note: the control probes use different token positions (sentence-final for question/statement, response-final for long/short, 100 examples each) than RIT/NIT collection points. A tighter control would probe a non-reflection distinction at the exact same positions. But the magnitude gap (chance vs >99%) makes positional artifacts an unlikely explanation.
> >
> > **On the "latent capability" reframing:**
> >
> > Footnote 1 now reads that whether self-reflection constitutes genuine metacognitive capability "remains further investigation and is beyond the scope of this paper." Appropriate and honest.
> >
> > The prefix-control experiments described in the response (showing that only coherent reflective content, not length-matched or style-matched controls, elicits reflection) are compelling but again not in the manuscript. If added, they would substantially strengthen Section 4.1.
> >
> > **On judge validation:**
> >
> > Appendix C now names Gemini-3-Flash and provides a validation table (Table 7): 25 positive examples from MATH500 with genuine reflection, 25 negative examples from fineweb-edu web text, near-perfect agreement across 3 runs. My concern remains: web text is a trivial negative. The actually hard boundary distinguishing genuine self-correction from extended deliberation that looks reflective but doesn't identify specific errors is untested. The downstream results indirectly suggest the judge works, but the validation itself doesn't probe the failure mode that matters for vector quality.
> >
> > **On the experiments only in the response letter:**
> >
> > Taking the reported numbers at face value, the HumanEval results (+6.7pp for Qwen2.5-7B, +3.2pp for DeepSeek-R1-7B) and sample-count ablation (works at any N from 25 to 500) are useful additions. The prefix-control table is well-designed. But I cannot evaluate results I can only see in a response letter they need to be in the manuscript with proper context, methodology description, and (for HumanEval) error bars.

---

> > > ### Author Response · Authors · 2026-05-28
> > >
> > > **Q: Add new experiments to the paper**
> > >
> > > A:
> > > We thank the reviewer for pointing this out.
> > > We have now incorporated the newly reported experiments into the manuscript rather than leaving them only in the response letter.
> > > Specifically, we added the dose-response reflection-rate analysis in Appendix E.1, the control-direction comparison in Appendix E.2, the prefix-control experiments in Appendix E.3, the HumanEval generalization results in Appendix E.4, and the sample-count ablation in Appendix E.5.
> > >
> > > **Q: Dose-response curve and functional effect of increased reflection**
> > >
> > > A:
> > > We have updated Fig. 6(a) in Appendix E to include both 1.5B and 7B models.
> > > The 7B models show the same qualitative trend as the 1.5B models: positive $\alpha$ increases judge-labeled self-reflection, while negative $\alpha$ suppresses it.
> > > This strengthens the evidence that SR steering modulates reflection frequency across model scales.
> > >
> > > The downstream results in Table 1 also provide supporting evidence for the second link in the reviewer’s proposed chain, namely that increased self-reflection can improve accuracy.
> > > SR enhancement improves performance on several reasoning tasks, while SR suppression often reduces performance or generation length.

---

> > > > ### Comment · Reviewer_YWp7 · 2026-06-02
> > > > **Thanks for incorporating everything into the manuscript the paper reads much more complete now, and the control-direction table in particular is exactly what was needed.**
> > > >
> > > > I have a few clarifying questions before I finalize things:
> > > >  * In Table 8, the caption mentions "three runs with different seeds" to explain the error bars. But Table 1 reports the same DeepSeek-R1-7B baseline as a point estimate under greedy decoding. What's actually varying across those three seeds? If it's the subset of examples used to construct the vector, just say so right now a reader will be confused about why the same model shows variance in one table but not the other.
> > > >
> > > >  * The AIME bootstrap CI for SR Enhanced on DeepSeek-R1-7B is +2.4 [+0.7, +6.9] that's noticeably right-skewed. It makes me wonder whether the improvement is coming from a couple of specific problems rather than a broad lift. Any intuition on whether the gains are concentrated or spread out?
> > > >
> > > >  * Something that caught my eye: the MATH-to-GPQA cosine similarity exceeds 0.9, but SR Enhanced actually hurts Qwen2.5-7B on GPQA Diamond (16.2 → 14.8). If the vectors are nearly the same direction, what's going wrong on GPQA specifically? Even a brief hypothesis would help the current "knowledge-intensive tasks" explanation is a bit hand-wavy.

---

> ### Author Response · Authors · 2026-06-03
>
> **Comment.** In Table 8, the caption mentions "three runs with different seeds" to explain the error bars. But Table 1 reports the same DeepSeek-R1-7B baseline as a point estimate under greedy decoding. What's actually varying across those three seeds? If it's the subset of examples used to construct the vector, just say so right now a reader will be confused about why the same model shows variance in one table but not the other.
>
> **Response.** Thank you for pointing this out. We have clarified in Tables 8 and 11 that these experiments do not use greedy decoding; instead, they use stochastic decoding with different random seeds to estimate run-level uncertainty. For steering-based methods, the seeds additionally affect the sampling of self-reflection pairs used to construct the steering vector, so each seed may produce a slightly different estimated direction and downstream performance. We have clarified this distinction in the captions of Tables 8 and 11 and in Appendix E.2 and E.6.
>
> **Comment.** The AIME bootstrap CI for SR Enhanced on DeepSeek-R1-7B is +2.4 [+0.7, +6.9] that's noticeably right-skewed. It makes me wonder whether the improvement is coming from a couple of specific problems rather than a broad lift. Any intuition on whether the gains are concentrated or spread out?
>
> **Response.** Since AIME contains relatively few examples, several problem-level changes can noticeably affect the bootstrap confidence interval and may contribute to the observed right-skew.
>
> To make this clearer, we added the per-run accuracies in Appendix E.6 / Table 11. SR Enhanced obtains 16, 18, 18, and 18 correct answers across the four runs, compared with 16, 16, 17, and 18 for Vanilla and 15, 16, 17, and 18 for BF. Thus, the gain is not driven by a single anomalous run, although it should not be interpreted as a uniform improvement across all problems.
>
> At the item level, the additional problems solved by SR Enhanced relative to BF are:
>
> - Run 1: Problem 14
> - Run 2: Problems 12 and 14
> - Run 3: Problem 12
> - Run 4: no additional problem
>
> We therefore interpret the AIME result conservatively. SR Enhanced mainly helps on problems where the model's initial trajectory is close enough that additional self-checking can correct an arithmetic error, a flawed intermediate derivation, or a premature conclusion. For harder AIME problems that are beyond the current capability of DeepSeek-R1-7B, simply enhancing self-reflection is not expected to solve the problem. In this sense, SR steering should be viewed as a mechanism for improving some self-check-correctable failures, not as a general solution to all difficult reasoning instances.
>
>
> **Comment.** Something that caught my eye: the MATH-to-GPQA cosine similarity exceeds 0.9, but SR Enhanced actually hurts Qwen2.5-7B on GPQA Diamond (16.2 → 14.8). If the vectors are nearly the same direction, what's going wrong on GPQA specifically? Even a brief hypothesis would help the current "knowledge-intensive tasks" explanation is a bit hand-wavy.
>
> **Response.** We have added a brief discussion of this issue on page 9, highlighted in purple. High MATH-to-GPQA cosine similarity does not imply that SR steering should have the same downstream effect on both tasks. Our interpretation is that the vector captures a broadly shared reflective state, but the utility of entering this state is task- and model-dependent.
>
> On MATH/AIME, many errors are caused by arithmetic mistakes or flawed intermediate derivations, so additional self-checking can correct the trajectory. In contrast, for Qwen2.5-7B on GPQA Diamond, we hypothesize that many failures stem from missing or uncertain domain knowledge rather than insufficient reflection. In such cases, enhancing reflection may cause the model to deliberate longer over incomplete or incorrect premises, amplify spurious reflection, or revise away from a correct initial answer. This explains why the same reflection direction can transfer geometrically while still producing negative task-level effects on GPQA.

---

### Decision · Action_Editor_z9ZR · 2026-06-17

**Recommendation:** Accept with minor revision

**Additional Comments:**

1.Calibration of claims about reflection mechanisms. The added prefix controls are helpful in ruling out simple length or style confounds, but they do not fully distinguish reflection-related pattern completion from stronger meta-cognitive interpretations. It would therefore be preferable to further soften claims and frame the findings more cautiously.

2.Hard-negative validation for the LLM judge. The LLM-as-a-judge pipeline is now better documented, but the current validation set still appears relatively easy. Adding hard negatives, such as long deliberative responses containing markers like wait without actual error identification or revision, would make the judge validation more convincing.

3.Correctness-conditioned ROSCOE-SS analysis. The rebuttal mentions a correctness-conditioned ROSCOE-SS analysis, but this analysis does not appear to be included in the revised manuscript.

**Audience:**

Yes

**Audience Explanation:**

This work focuses on whether self-reflection is a novel behavior induced by RLVR or already present during pretraining, and how to control self-reflection to affect reasoning performance and efficiency. This is a timely and actively discussed topic in the LLM community and the findings are directly useful to researchers studying LLM reasoning and the internal mechanisms behind reasoning behavior.

**Claims And Evidence:**

Yes

**Claims Explanation:**

This work studies the emergence and controllability of self-reflection in LLMs. It proposes Reflection-Inducing Probing to test whether pretrained models can exhibit reflective behavior and constructs a self-reflection vector to steer reflective behavior during inference.

Accurate and Clear Evidence

1.The main empirical claims are supported by clear and convincing evidence. The steering effect itself is demonstrated across multiple model families and benchmarks. In Table 1, SR Enhanced generally improves Pass@1, while SR Suppressed reduces response length with relatively small accuracy degradation for fine-tuned models.

2.The evidence for representation-level separability is also clear. The paper shows reflection and non-reflection are well separated in hidden states in Figure 1, 4 and 5.

3.The paper also provides useful ablations. Figure 2 shows that moderate positive steering improves performance and that middle-layer injection is most effective.

---

> ### Author Response · Authors · 2026-07-17
>
> We thank the editor for the constructive comments. We have revised the manuscript accordingly, as detailed below.
>
> ### 1. Calibration of claims about reflection mechanisms
>
> We agree that our evidence does not distinguish reflection-related pattern completion from stronger metacognitive interpretations. We have therefore expanded Footnote 1 to clarify that our claims concern reflective behavior and its internal representations, rather than metacognition per se. We have also revised the corresponding discussion in Section 3.2 to frame the interpretation of the results more cautiously.
> >However, whether self-reflection should be interpreted as genuine metacognitive capability requires further investigation and is beyond the scope of this paper. Our claims throughout are therefore about reflective behavior and its internal representations, rather than about metacognition per se.
>
> > For this reason, we do not define self-reflection using a fixed set of tokens alone. Instead, we use an LLM-as-a-judge~\citep{gu2024survey} to evaluate complete responses and determine whether they exhibit self-reflective behavior.
>
> ### 2. Hard-negative validation for the LLM judge
>
> Following the editor's suggestion, we constructed a hard-negative validation set of 25 examples by prompting a separate LLM (`GPT5.5`) to answer the first 25 MATH-500 problems with long deliberative responses containing explicit reflection markers, such as “wait” and “recheck,” but without performing any actual error identification or revision. All 25 examples were manually verified by the authors to confirm the absence of genuine reflective revision.
>
> Because our downstream analysis uses binary reflection labels, a hard negative is counted as correctly classified if the judge assigns either `SURFACE_REFLECTION` or `NO_REFLECTION`. Across three repeated runs, the judge correctly rejected all 25 hard negatives, indicating that it distinguishes surface-level reflection markers from genuine revision rather than relying only on keyword presence.
>
> We have expanded Appendix C to include the hard-negative construction procedure, the generation prompt, a representative example, and the extended judge-validation results. The complete set of hard-negative examples is also included as a JSON file in the supplementary material.
>
> ### 3. Correctness-conditioned ROSCOE-SS analysis
>
> We have added the correctness-conditioned ROSCOE-SS analysis to Appendix E.8 of the revised manuscript. This appendix reports ROSCOE-SS separately for examples with correct and incorrect final answers under the Vanilla, SR Enhanced, and SR Suppressed settings, together with a brief discussion of the motivation and interpretation of the results.